# Data-driven network alignment

Shawn Gu[1,2,3], Tijana Milenković[1,2,3]*

**1** Department of Computer Science and Engineering, University of Notre Dame, Notre Dame, IN, United States of America, **2** Eck Institute for Global Health, University of Notre Dame, Notre Dame, IN, United States of America, **3** Center for Network and Data Science, University of Notre Dame, Notre Dame, IN, United States of America

* tmilenko@nd.edu

**Data Availability Statement:** The software and data are available at http://www.nd.edu/~cone/TARA/.

**Funding:** This work is supported by the Air Force Office of Scientific Research Young Investigator Research Program (https://www.nsf.gov/funding/pgm_summ.jsp?pims_id=503214) FA9550-16-1-

## Abstract

In this study, we deal with the problem of biological network alignment (NA), which aims to find a node mapping between species' molecular networks that uncovers similar network regions, thus allowing for the transfer of functional knowledge between the aligned nodes. We provide evidence that current NA methods, which assume that topologically similar nodes (i.e., nodes whose network neighborhoods are isomorphic-like) have high functional relatedness, do not actually end up aligning functionally related nodes. That is, we show that the current topological similarity assumption does not hold well. Consequently, we argue that a paradigm shift is needed with how the NA problem is approached. So, we redefine NA as a data-driven framework, called TARA (data-driven NA), which attempts to learn the relationship between topological relatedness and functional relatedness without assuming that topological relatedness corresponds to topological similarity. TARA makes no assumptions about what nodes should be aligned, distinguishing it from existing NA methods. Specifically, TARA trains a classifier to predict whether two nodes from different networks are functionally related based on their network topological patterns (features). We find that TARA *is* able to make accurate predictions. TARA then takes each pair of nodes that are predicted as related to be part of an alignment. Like traditional NA methods, TARA uses this alignment for the across-species transfer of functional knowledge. TARA as currently implemented uses topological but not protein sequence information for functional knowledge transfer. In this context, we find that TARA outperforms existing state-of-the-art NA methods that also use topological information, WAVE and SANA, and even outperforms or complements a state-of-the-art NA method that uses both topological and sequence information, PrimAlign. Hence, adding sequence information to TARA, which is our future work, is likely to further improve its performance. The software and data are available at http://www.nd.edu/~cone/TARA/.

## Introduction

### Background and motivation

Networks are commonly used to model complex real-world systems in many domains, including computational biology. Protein-protein interaction (PPI) networks are a widely studied

0147, awarded to TM, and the National Science Foundation Faculty Early Career Development Program (https://www.nsf.gov/funding/pgm_summ.jsp?pims_id=503214) CCF-1452795, awarded to TM. The funders had no role in study design, data collection and analysis, decision to publish, or preparation of the manuscript.

**Competing interests:** The authors have declared that no competing interests exist.

type of biological networks, and they are often used to model cellular functioning. In such networks, nodes are proteins and edges are PPIs. While biotechnological advancements have made PPI network data available for many species [1–4], functions of many proteins in many of these species remain unknown. One way to uncover these functions is by transferring biological knowledge from a well-studied species to a poorly-studied one. Genomic sequence alignment can be used for this purpose, but one drawback of doing so is that it does not consider the interactions between proteins (which are ultimately what carry out function). So, network alignment (NA) can be used in a complementary fashion to predict what sequence alignment alone cannot [5–10]

NA aims to find a node mapping between the compared networks that uncovers regions of high topological (and often sequence) similarity. This is closely related to the subgraph isomorphism, or subgraph matching, problem, in which the goal is to find a node mapping such that one network is an exact subgraph of the other [11]. However, NA is more general in that it aims to find the best "fit" of one network to the other, even if the first is not an exact subgraph of the second. Still, existing NA methods borrow ideas from the subgraph isomorphism problem. Prominently, they use the notion of topological similarity, which quantifies how close to isomorphic two nodes' extended neighborhoods are. The underlying subgraph isomorphism problem is NP-hard [11]. So, topological similarity is measured heuristically [12, 13]. While different existing NA methods typically use mathematically different heuristics [14, 15], the notion of topological similarity that they aim to capture is the same and as mentioned above: how isomorphic-like their extended network neighborhoods are. Then, these methods align nodes that are topologically similar to each other to try to find the best "fit" between the compared networks. Analogous to sequence alignment, functional knowledge can be transferred between conserved (aligned) network, rather than (just) sequence, regions. Note that while our focus is on computational biology, NA is applicable to many other domains as well [8] (e.g., machine translation in natural language processing [16], identity matching across different social media platforms [17–19], or visual feature matching in computer vision [20]).

NA can be categorized in several ways. First, like sequence alignment, NA can be local or global [7, 10]. Local NA methods aim to find highly conserved regions across the compared networks, usually leading to such regions being small, while global NA methods try to find a node mapping that maximizes the overall similarity of the compared networks, usually leading to such regions being large. As a result, local NA methods generally find alignments of higher functional quality than global NA methods, and global NA methods generally find alignments of higher topological quality than local NA methods [7]. However, while local NA methods can still produce alignments of reasonably high topological alignment quality (just usually not as high as those of global NA methods), local NA methods, and thus also global NA methods, usually do not result in alignments of high functional quality [7]. So, there is a need for improving both types of NA. But because global NA has received more attention recently, we focus on the problem of global NA in this paper.

Second, NA can be pairwise or multiple [6, 10]. Pairwise NA methods align exactly two networks, while multiple NA methods align more than two networks at once. Because multiple NA methods are more computationally complex than pairwise NA methods [21], and because they are also less accurate than current pairwise NA methods, [22], we focus on the problem of pairwise NA in this study.

Third, NA can be categorized based on whether the output is a one-to-one or many-to-many alignment. In a one-to-one alignment, a node $u$ in a network $G$ can be aligned to exactly one node $v$ in another network $H$, and that particular node $v$ in network $H$ cannot be aligned to any other node in network $G$. On the other hand, in a many-to-many alignment, a node $u$ in a network $G$ can be aligned to multiple nodes, including $v$, in another network $H$, and that

particular node *v* in *H* can be aligned to other nodes in *G* in addition to node *u*. While we propose a many-to-many NA approach in this study (see below), we evaluate against both one-to-one and many-to-many NA methods.

Fourth, NA methods can be divided into those that consider topological information from the input networks, aligning nodes if their topologies, i.e., network neighborhoods, are similar, and those that additionally use external, non-topological information in the form of anchor links between nodes from the different networks prior to aligning the networks. For example, in the biological domain, sequence similarities between proteins are typically used to link proteins across molecular networks of different species. Or, in the social domain, known identities of users are typically used to link users' accounts across different social networks corresponding to different online media platforms. Alignments are then built around these anchor links, while also accounting for topological similarities between nodes across the different networks (like the first method type). We propose an NA method that does not use anchor links (see below), but we evaluate against both types of methods.

One major issue of current NA methods, no matter what NA method category they belong to, is that regardless of what measure of topological similarity they use, their aligned nodes often do not correspond to nodes that should actually be mapped, i.e., that are functionally related. For example, when comparing PPI networks of different species, aligned nodes (proteins) do not correspond to proteins that perform the same biological function—in other words, a correlation between topological alignment quality and functional alignment quality has yet to be observed [7, 10, 13, 23, 24].

In this study, we argue that this is because the traditional assumption of existing NA methods, namely that topologically similar nodes (with isomorphic-like neighborhoods) have high functional relatedness and should thus be mapped to each other, does not hold. This may be caused by several factors, as follows.

First, current PPI network data are highly noisy, with many missing and spurious PPIs (and even proteins) [25]. This alone can cause mismatches between proteins' topological similarity and their functional relatedness. For example, if a set of three proteins that are all linked to each other via PPIs (i.e., a triangle) is in reality fully evolutionary conserved (i.e., functionally related) between two species, then the two triangles in the two species are topologically similar. But say that one of the three PPIs that actually exists in reality is missing in exactly one of the two species' current PPI networks due to data noise. Then, it is a 3-node path in that species that should be aligned to a triangle in another species in order to identify the functional match. That is, the functionally related regions are now topologically dissimilar due to the data noise.

Second, even when PPI network data become complete, the traditional assumption of topological similarity is unlikely to hold due to biological variation between species. Namely, molecular evolutionary events such as gene duplication, deletion, or mutation may cause PPI network topology to differ across species' evolutionary conserved (i.e., functionally related) network regions. Even for protein sequence alignments, pairwise sequence identity as low as 30% is sufficient to indicate evolutionary conservation (i.e., homology) for 90% of all protein pairs [26]. So, one can perhaps expect evolutionary conserved PPI networks of different species to be as topologically dissimilar.

Third, there could be additional factors that have yet to be discovered.

Regardless of the causes, our study is the first to provide actual evidence that the traditional topological similarity assumption does not hold. Briefly, we investigate whether functionally related nodes are indeed topologically similar in two tests: on synthetic networks and on real-world PPI networks of different species. In the process, we consider multiple prominent measures of topological similarity. As discussed in Section "Results—Topological similarity versus

functional relatedness", we find that functionally related nodes are only marginally more similar (for all considered measures of topological similarity) to each other than at random. This means that aligning topologically similar nodes, as existing NA methods do, has only a marginally higher chance of aligning functionally related nodes than functionally unrelated nodes.

## Our contributions

Because the assumption of existing NA methods that topologically (and sequence) similar nodes should be aligned (i.e., are functionally related) does not hold, we propose a new paradigm for NA. Namely, we aim to redefine NA as a data-driven framework, which attempts to learn from the data what kind of topological relatedness corresponds to functional relatedness, without assuming that topological relatedness means topological similarity. So, regardless of whether the traditional topological similarity assumption fails due to noisy data, biological variation, or something else, we hypothesize that topological relatedness can better capture functional relatedness than topologically similarity can. As topological relatedness and topological similarity are important concepts for understanding our paper, we illustrate their difference in Fig 1 and explain it below.

Suppose that we are aligning PPI networks of two different species, where for simplicity, only parts of the whole networks are shown in Fig 1. Also, suppose that color corresponds to the function that a node performs, in this case the "purple" function or the "orange" function. An NA method based on topological similarity will produce an alignment with low functional quality on our example networks (Fig 1). Such a method will align nodes $d$, $e$, $f$, and $g$ in species 1 to nodes 1, 2, 3, and 8 in species 2 (Fig 1(a)) because each set of nodes forms the same subgraph: a square with a diagonal (square-with-diagonal). However, the species 1 nodes perform the "orange" function, while the species 2 nodes perform the "purple" function—the nodes are not functionally related. On the other hand, an NA method based on topological relatedness will produce an alignment with high functional quality on our example networks. This is because such a method will learn that 3-node paths in species 1 should be aligned to square-with-diagonals in species 2, since the 3-node path consisting of nodes $a$, $b$, and $c$ in species 1 performs the same function ("purple") as the square-with-diagonal consisting of nodes 1, 2, 3, and 8 in species 2; and that square-with-diagonals in species 1 should be aligned to squares in species 2, since the square-with-diagonal consisting of nodes $d$, $e$, $f$, and $g$ in species 1 performs the same function ("orange") as the square consisting of nodes 4, 5, 6, and 7 in species 2. Using

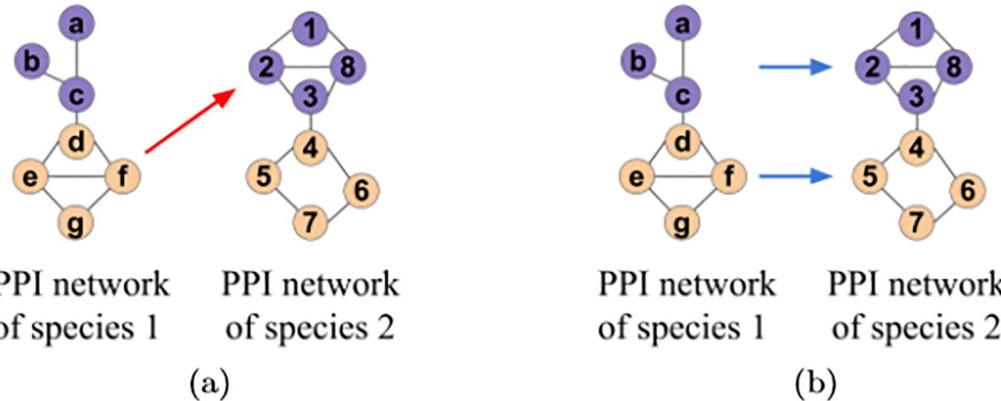

**Fig 1. Illustration of (a) the existing notion of topological similarity versus (b) our new notion of topological relatedness.**

these learned patterns, the method will try to align the rest of the nodes between the networks (not shown in the figure), transferring the functions of 3-node paths to square-with-diagonals, and of squares-with-diagonals to squares. In essence, noisy data or evolutionary events can be captured by topological relatedness but not topological similarity.

With our new notion of topological relatedness, we make no assumptions about what nodes should be aligned, distinguishing us from existing NA methods. Specifically, as a proof-of-concept methodological solution to test our new paradigm, we train a supervised classifier that, given a topological feature vector (i.e., low-dimensional embedding) of a node pair, learns from the (training) data when nodes are functionally related and when they are not. Note that because state-of-the-art topological features in the field of NA rely on graphlets [6, 10], we use graphlet-based feature vectors in our new framework. Importantly, we do not use any anchor links between nodes of different networks in order to calculate the feature vector of a node pair, unlike many existing methods. Then, we use pairs (from the testing data) whose nodes are predicted to be functionally related to build an alignment. In other words, we consider two nodes to be topologically related if they are predicted to be functionally related, and our framework aligns such nodes, unlike existing methods that align topologically similar nodes. Of course, we make predictions only for node pairs that are not in the training data, which avoids any circular argument. So, we convert the NA problem into the problem of across-network supervised protein functional classification. While established supervised versions of many problems do exist, supervised NA has barely been studied before. We refer to the entire framework described above as TARA (data-driven network alignment).

TARA is a global, pairwise, and many-to-many method that does not use sequence similarity-based anchor links. We evaluate TARA against three state-of-the-art NA methods that are as similar as possible to TARA in terms of their algorithmic design or output, namely against WAVE [27], SANA [28], and PrimAlign [29]. Specifically, just like TARA, WAVE and SANA are global and pairwise, do not use anchor links, and furthermore are also graphlet-based. The only difference is that WAVE and SANA are one-to-one, unlike TARA. So, we also analyze PrimAlign, which is many-to-many and also global and pairwise, like TARA. Unlike TARA, PrimAlign does use anchor links in the form sequence similarities between networks. We evaluate each method on both synthetic (geometric and scale-free) and real-world (yeast and human PPI) networks.

Overall, we find that TARA is able to accurately learn what kind of topological relatedness corresponds to functional relatedness, and that TARA is able to predict the functions of proteins more accurately or in a complementary fashion compared to the existing NA methods, even those that use both topological and sequence information, mostly at lower running times. Thus, there is a need for introducing our new data-driven approach.

## Related work

Traditional biological NA methods typically consist of two algorithmic components. First, the similarity between pairs of nodes is computed with respect to topology, sequence, or both. Second, an alignment strategy identifies alignments that maximize the similarity between aligned node pairs and the amount of conserved edges (intuitively, alignments should preserve interactions). There are two common types of alignment strategies, as follows.

One is seed-and-extend, where first two highly similar nodes are aligned, i.e., seeded. Then, the most similar of the seed's neighboring nodes (or simply neighbors), the neighbors of the seed's neighbors, etc. are aligned. This continues until all nodes of the smaller of the two networks are aligned (until a one-to-one node mapping between the two networks is produced).

WAVE [27] is a state-of-the-art seed-and-extend alignment strategy that works the best under a graphlet-based topological similarity measure.

The other type of alignment strategy is a search algorithm. Here, instead of aligning node by node like a seed-and-extend method, the solution space of possible alignments is explored, and the one that scores the highest with respect to some objective function is returned. This objective function typically tries to maximize the overall node similarity and the number of conserved edges. SANA [28] is a state-of-the-art search algorithm-based method. Specifically, it uses simulated annealing to search through possible one-to-one alignments, and works the best under an objective function that maximizes the overall graphlet-based topological similarity as well as the number of conserved edges.

On the other hand, PrimAlign [29] is a method with an alignment strategy that does not strictly belong to one of the above two categories. PrimAlign models the network alignment problem as a Markov chain where every node from one network is linked to some or all nodes in the other network with some scores; for PPI networks, these scores can be sequence similarities. In other words, PrimAlign makes use of anchor links between networks. The chain is then repeatedly transitioned until convergence, redistributing the across-network link scores using a PageRank-inspired algorithm. Those links that are above a certain threshold are taken as the alignment. As a result, PrimAlign outputs a many-to-many alignment, where a protein from one network may be aligned to many proteins in the other.

A method called MUNK has appeared recently [30]. Like PrimAlign, MUNK also relies on sequence-based anchor links (specifically, homologs) between two networks, but unlike PrimAlign, MUNK uses a matrix factorization approach to embed the nodes into a low dimensional space. Then, it uses these embeddings to calculate similarities between pairs of nodes, and employs the Hungarian algorithm on these similarities to generate an alignment. In preliminary analyses of MUNK on our data, we found that the similarity scores were not able to distinguish between functionally related and functionally unrelated nodes. This, combined with the fact that MUNK appeared after our study has been completed, is why we do not pursue it further.

The above methods do not use any functional (i.e., Gene Ontology (GO) [31] information in the alignment process, unlike TARA. However, one method, DualAligner [32], does use such information, albeit in a different way than TARA. Given two PPI networks where some of the proteins are annotated with GO terms, DualAligner first forms "function-constrained subgraphs" (connected subgraphs sharing a GO term) in each network. Then, it tries to align subgraphs of the same function across networks. Next, it aligns proteins within these subgraphs that are topologically and sequence similar. Finally, it uses a seed-and-extend strategy around these aligned pairs to match unannotated proteins. However, more recent, state-of-the-art methods have appeared since DualAligner, including WAVE, SANA, and PrimAlign, which is why we do not consider DualAligner in this study.

All of the above methods are unsupervised. That is, they assume that topologically similar nodes are functionally related. Of course, many other such methods exist [10]. However, in the WAVE, SANA, and PrimAlign studies, the three methods were shown to outperform a number of the previous NA methods including AlignMCL [33], AlignNemo [34], CUFID [35], HubAlign [36], IsoRankN [37], L-GRAAL [38], MAGNA [39], MAGNA++ [40], MI-GRAAL [41], NETAL [42], NetCoffee [43], NetworkBLAST [44], PINALOG [45], and SMETANA [46]. In turn, these methods were shown to outperform GHOST [13], IsoRank [47], NATALIE [48], PISwap [49], and SPINAL [50]. So, the fact that WAVE, SANA, and PrimAlign are state-of-the-art, coupled with the fact that they are the most directly comparable to TARA (in terms of algorithmic design or output), is why we focus on them.

In addition, two supervised methods do exist, IMAP and MEgo2Vec, as follows.

IMAP [51] is an NA method that incorporates supervised learning, but in a different way than what we propose. First, IMAP requires an (unsupervised) alignment between two networks as input. Then, it obtains a topological feature vector for each node pair. Node pairs that are aligned form the positive class, and node pairs that are not aligned are sampled to form the negative class. Then, IMAP uses this data to train a linear regression classifier. After training, the data is passed through the classifier again in order to assign a score to every node pair. These scores are used in a matching algorithm (e.g., Hungarian or stable marriage) to form a new alignment, which is then given back as input into the method. This process is repeated for a set number of iterations—in general, it is shown that these iterations improve alignment quality. However, IMAP still makes the assumption that topologically similar nodes should be mapped to each other, meaning it still suffers from the issues of other NA methods. TARA on the other hand learns from the data what kind of topologically related nodes should be mapped to each other. We did attempt to test IMAP in this study, but the code was not available, and when we tried to implement it ourselves, we could not get the method to work (i.e., we were not able to reproduce results from the IMAP study).

MEgo2Vec [52] proposes a framework to try to match user profiles across different social media platforms. Using graph neural networks and natural language processing techniques to obtain features of pairs of profiles from different platforms, MEgo2Vec then trains a classifier to predict whether two profiles correspond to the same person. However, because MEgo2Vec uses text processing techniques to match users' names, affiliations, or research interests (in addition to network topological information), it is not directly suitable for matching proteins across PPI networks. Unlike MEgo2Vec, TARA relies solely on topological information (although it can also use external, e.g., sequence, information, this is out of the scope of this study).

There also exists a variety of methods that aim to predict the function of proteins *within a single* PPI network using techniques such as guilt-by-association, clustering, or classification [53, 54]. While this is a valuable research area, we are interested in a different problem—that of *across-network* protein function prediction. As such, we do not consider single-network methods in this study.

Lastly, there exist methods that aim to predict protein function without using any PPI network information. A variety of approaches entered in the Critical Assessment of Functional Annotation (CAFA) challenge [55] fall under this category. For example, the most recent top performing method, GOLabeler [56], uses a combination of protein sequence, amino acid, structural, and biophysical information, in order to predict GO term annotations of proteins. However, these kinds of *non-network* approaches are out of the scope of our *network-focused* study.

## Materials and methods

### Data

Like many NA methods do, we evaluate TARA on network sets with known node mapping (networks generated from different graph models and their randomly perturbed counterparts) and a network set with unknown node mapping (yeast and human PPI networks).

**Network sets with known node mapping.** We use two network sets with known node mapping, generated from two network (i.e., random graph) models: 1) geometric random graphs [57] and 2) scale-free networks [58]. Because these two models have distinct network topologies [59], we can test the robustness of our results to the choice of model. For a given model, we create a network with 1,000 nodes and 6,000 edges, and then generate five instances of $x$% random perturbation (i.e., we randomly delete $x$% of the edges and then randomly add

the same number of edges back), varying $x$ to be 0, 10, 25, 50, 75, and 100. Because only edges differ between the original network and a randomly perturbed counterpart, we know the correct node mapping, and pairs in this mapping can be considered to be "functionally" related.

**Network set with unknown node mapping.**    We use the PPI networks of yeast (5,926 nodes and 88,779 edges) and human (15,848 nodes and 269,120 edges) analyzed by the PrimAlign study [29], obtained from BioGRID [60]. Because we do not know the true node mapping between these networks, we rely on GO annotations to measure the functional relatedness between proteins (discussed below). We accessed the GO data in November 2018.

## TARA: Data-driven network alignment

Recall that TARA trains, on a portion of the data, a supervised classifier using topological relatedness-based feature vectors of node pairs and their labels (whether the nodes in a given pair are functionally related or not). Then, it aims to predict, on the remainder of the data, if a node pair is functionally related, creating an alignment out of all pairs predicted as such. Finally, this alignment can be used in a protein function prediction framework. Below, we describe how a topological relatedness-based feature vector of a node pair is extracted (subsection "Topological relatedness of a node pair."), how the classifier is trained and evaluated on each of the network sets (subsections "TARA for a network set with known/unknown node mapping."), and how the protein function prediction framework works (subsection "TARA as an NA framework for protein function prediction.").

**Topological relatedness of a node pair.**    We quantify topological relatedness using the notion of graphlets. Graphlets are Lego-like building blocks of complex networks, i.e., small subgraphs of a network (a path, triangle, square, etc.). In this study, we consider up to 5-node graphlets. They can be used to summarize the extended neighborhood of a node as follows. For each node in the network, for each topological node symmetry group (formally, automorphism orbit), one can count how many times a given node touches each graphlet at each of its orbits. The resulting counts for all graphlets/orbits are the node's *graphlet degree vector* (GDV) [12], which has a length of 73 for up to 5-node graphlets. Then, to obtain the feature of a node pair, we simply take the absolute difference of the nodes' GDVs (GDVdiff). Note that in addition to GDVdiff, we also tested appending the nodes' GDVs together (GDVappend), and a weighted difference of the nodes' GDVs based on the GDV similarity [12] calculation (GDVsim). However, the GDVdiff outperformed GDVappend, and while it obtained similar results to GDVsim, GDVdiff is mathematically simpler. As such, we only focus on GDVdiff, as calculated in Algorithm 1.

**Algorithm 1** Extracting the "GDVdiff" feature vector of a node pair. Given two networks $G_1$ and $G_2$, and a node pair $s_{ij} = (v_{1_i}, v_{2_j})$, define a function $g : (s_{ij}, G_1, G_2) \to \mathbb{R}^{73}$, computed as follows. For notations and their meanings, see Table 1.
1: let $\mathbf{a}_{1_i} = f(v_{1_i}, G_1)$
2: let $\mathbf{a}_{2_j} = f(v_{2_j}, G_2)$
3: **return** $abs(\mathbf{a}_{1_i} - \mathbf{a}_{2_j})$

**TARA for network sets with known node mapping.**    First, in order to see whether functional relatedness can even be predicted from topological relatedness, we evaluate our classifier using 10-fold cross-validation; if not, further study would be pointless. To do so, we start by creating a dataset that is balanced between the positive class (node pairs that are known functionally related) and the negative class (node pairs that are not currently known to be functionally related). But, because there are many more node pairs in the negative class, we undersample them to match the positive class in size, a common technique when dealing with class imbalance [32]. The process for creating one balanced dataset is outlined in Algorithm 2.

**Table 1. Table of notations and their meanings.**

| Notation | Meaning |
|---|---|
| $G_i$ | Network $i$ |
| $V_i$ | Node set of network $i$ |
| $E_i$ | Edge set of network $i$ |
| $|S|$ | Size of a set $S$ |
| $v_{i_j}$ | $j$th node of network $i$, i.e., $v_{i_j} \in V_i$ |
| $f(\_, G_i)$ | $f : (v_{i_j}, G_i) \rightarrow \mathbb{R}^{73}$, a function that returns the GDV of node $v_{i_j} \in V_i$ |
| $g(\_, G_i, G_j, d)$ | Defined in Algorithm 1 |
| $bal(G_i, G_j, R, R', d)$ | Defined in Algorithm 2 |
| $abs(\mathbf{x})$ | Element-wise absolute value of a vector $\mathbf{x}$ |
| `random.sample`$(S, n, d)$ | From set $S$, randomly sample $n$ elements without replacement, based on random seed state $d$ |
| $U \times V$ | Cartesian product of sets $U$ and $V$ |
| $getAln(G_i, G_j, R, R', d, y)$ | Defined in Algorithm 3 |

Then, given this balanced dataset, we split the data into training and testing sets. We sample 90% of the data to become the training set (ensuring balanced class sizes), and so the remaining 10% becomes the testing set. For 10-fold cross-validation, we take 10 stratified samples so that each data instance appears in exactly one of the ten testing sets, resulting in 10 "folds". Given a fold, we then train a logistic regression classifier using the GDVdiff feature for a node pair to predict whether the given two nodes are functionally related, and evaluate this classifier using the accuracy and area under receiver operating characteristic curve (AUROC). For each score, we average over the 10 folds. We also repeat the undersampling 10 times to ensure any outcome is unlikely due to how we sample the negative class. So, we obtain 10 balanced datasets, and thus 10 accuracy and 10 AUROC scores, and for each measure we average the 10 scores. We repeat this process for every random perturbation amount. Note that we also tested Naive Bayes, decision tree, and simple neural network classifiers; trends were qualitatively similar, but logistic regression gave the best results. As such, we focus on logistic regression.

**Algorithm 2** Creating a balanced dataset. Given two networks $G_1$ and $G_2$, the set of conditions $R$ that a node pair needs to satisfy to be considered functionally related, the set of conditions $R'$ that a node pair needs to satisfy to be considered not functionally related, and random seed state $d$, define a function $bal : (G_1, G_2, R, d) \rightarrow (S_1, S'_2)$ such that $S_1 \subset V_1 \times V_2$ is a set of node pairs satisfying $R$ (functionally related) and $S'_2 \subset V_1 \times V_2$ is an equally sized set satisfying $R'$ (not functionally related). For notations and their meanings, see Table 1.

```
1: let S₁ be the set of node pairs between G₁ and G₂ satisfying R, i.e.,
the set of functionally related node pairs.
2: let S₂ be the set of node pairs between G₁ and G₂ satisfying R',
i.e., the set of functionally unrelated node pairs.
3: S'₂ = random.sample(S₂, |S₁|, d)
4: return(S₁, S'₂)
```

Second, we analyze the amount of data needed to train a good classifier, since only a small amount of data may be available in many real-world applications. For each network model, for each random perturbation amount, we obtain 10 balanced datasets using the same process as above. Then, for a given balanced dataset, we split the data such that $y$% goes into the training set and the remaining $(100 - y)$% goes into the testing set, still keeping the class balance in both the training and testing sets, varying $y$ from 10 to 90 in increments of 10. For a given value of $y$, i.e., for what we call a $y$ percent training test, we randomly create 10 instances of

this training and testing split, resulting in 10 accuracy and 10 AUROC scores, and for each measure we average results to ensure the outcomes are not due to the how we select the instances. Note that if $y = 90$ and we were to take stratified samples instead of fully random samples, we would be performing 10-fold cross-validation as above. Finally, we average over all 10 balanced datasets to ensure the outcomes are unlikely due to how we sample the negative class.

**TARA for a network set with unknown node mapping.** Since we do not know the node mapping between yeast and human PPI networks, we must define functional related-ness in a different way compared to for network sets with known node mapping. We use GO annotations to do this. Specifically, if a yeast-human protein pair shares at least $k$ biological process (BP) GO terms in which the protein-GO term annotations were experimentally inferred (i.e., if a given annotation has one of the following evidence codes: EXP, IDA, IPI, IMP, IGI, IEP), then we say the pair is functionally related. We vary $k$ from 1 to 3. This gives us three sets of ground truth data, which we refer to as *atleast1-EXP*, *atleast2-EXP*, and *atleast3-EXP*. Also, no matter what $k$ is, we define a functionally unrelated pair as a pair sharing no GO terms.

Then, for a given $k$, functionally related pairs form the positive class, and functionally unre-lated pairs form the negative class. Once again because there are many more negative pairs, we take 10 samples that match the size of the positive pairs to create 10 balanced datasets and aver-age over them. Again we use GDVdiff as the feature under logistic regression.

We again perform each of (i) 10-fold cross-validation and (ii) $y$ percent training tests on the 10 balanced datasets, just like before, except that in the percent training tests, we now only per-form the $y$% training/testing split once instead of 10 times for simplicity, since we find the training/testing split does not significantly change the results.

**TARA as an NA framework for protein function prediction.** In addition to 10-fold cross-validation and $y$ percent training tests (on each of networks with known and unknown node mapping), we evaluate TARA in a third test—that of protein function prediction. This is an important downstream task of NA. We evaluate TARA in this context as follows. For a given set of ground truth data atleast $k$-EXP, we keep only GO terms that annotate at least two yeast proteins and at least two human proteins; without this constraint, it is impossible for the framework (described below) to make predictions for the GO term. Then, for a given percent training test $y$, we train TARA and make predictions on the remaining testing data. Every pair that is predicted to be in the positive class is added to an alignment. We outline the process in Algorithm 3 This alignment, as well as alignments of existing methods that we evaluate against, is then put through the protein function prediction framework proposed by [7], which we summarize as follows. For each protein $u$ in the alignment (that is annotated by at least $k$ GO term(s)), we hide $u$'s true GO term(s). Then, for each GO term $g$, we determine if the align-ment is statistically significant with respect to $g$. This is done by calculating if the number of aligned node pairs in which the aligned proteins share $g$ is significantly high ($p$-value less than 0.05 according to the hypergeometric test [7]). After repeating for all applicable proteins and GO terms, we obtain a list of predicted protein-GO term associations. From this list we can calculate the precision and recall of the predictions.

**Algorithm 3** Generating an alignment. Given two networks $G_1$ and $G_2$, the set of conditions $R$ that a node pair needs to satisfy to be considered functionally related, the set of conditions $R'$ that a node pair needs to satisfy to be considered not functionally related, random seed state $d$, and a $y$ percent training amount, return an alignment of $G_1$ and $G_2$. For notations and their meanings, see Table 1.

```
1: Initialize set A to be empty.
2: let (Sp, Sn) = bal(G1, G2, R, R', d), as computed by Algorithm 2.
```

```
3: let Tr_p = random.sample(S_p, ⌊y|S_p|/100⌋, d).
4: let Te_p = S_p\Tr_p
5: let Tr_n = random.sample(S_n, ⌊y|S_n|/100⌋, d).
6: let Te_n = S_n\Tr_n.
7: let Tr = Tr_p ∪ Tr_n be the training data.
8: let Te = Te_p ∪ Te_n be the testing data.
9: Train a predictive function, LogReg : ℝ^73 → {0,1}, with logistic
regression, on Tr, where the feature vector of node pair s_ij ∈ Tr is
given by g(s_ij, G_1, G_2), and the label of s_ij is {1 if s_ij ∈ S_p, 0 if
s_ij ∈ S_n}.
10: for each s_ij ∈ Te do
11:    let x_ij = g(s_ij, G_1, G_2)
12:    if LogReg(x_ij) = 1 then
13:       A.add(s_ij)
14:    end if
15: end for
16: return A
```

For example, if $G_1$ is the yeast PPI network, $G_2$ is the human PPI network, $R$ is the set of conditions for the atleast3-EXP ground truth dataset, $R'$ is the set of conditions for a protein pair to be considered not functionally related (i.e., shared no GO terms of any kind), $d$ is 0, and $y$ is 90%, then $getAln(G_1, G_2, R, R', d, y)$ returns the alignment between yeast and human generated by a classifier trained on functionally related protein pairs defined by R, functionally unrelated pairs defined by R', using a random seed state of 0 for sampling, and 90% of the data for training.

While in traditional NA evaluations every GO term available is considered, some GO terms may be redundant or too general. Recent work has suggested that taking the frequency of GO terms (how many proteins a GO term annotates out of all proteins analyzed) into account can deal with these issues [62]; intuitively, less frequent means more informative. However, because there is no hard definition for what makes a GO term rare enough, we consider three thresholds:

- All GO terms (i.e., ALL); this corresponds to traditional NA evaluation.

- GO terms that appear 50 times or fewer (i.e., threshold of 50).

- GO terms that appear 25 times or fewer (i.e., threshold of 25).

For a given GO term rarity threshold, we filter out all GO terms that do not satisfy the threshold. Then, for each atleast $k$-EXP ground truth dataset (see above), we only consider proteins that share at least $k$ GO terms from the filtered list to be functionally related (keep in mind that for proteins to be considered functionally unrelated, they still must share no GO terms, regardless of rarity). For example, atleast1-EXP at the 50 GO term rarity threshold considers proteins that share at least one experimentally inferred biological process GO term, such that each GO term annotates 50 or fewer proteins (out of all proteins from the yeast and human PPI networks we analyze), to be functionally related. In total, we now have nine "ground truth-rarity" datasets, resulting from combinations of the three atleast $k$-EXP ground truth datasets and the three GO term rarity thresholds. Then, for each of these nine datasets, we train and test TARA on protein pairs satisfying the conditions (i.e., being in the atleast $k$-EXP ground truth dataset at the given GO term rarity threshold), and evaluate the resulting alignment using the protein function prediction framework described above. Also, in order to fairly compare TARA to all existing NA methods, we evaluate the existing methods' alignments with respect to these nine ground truth-rarity datasets. In this way, we can test the effect

of both $k$ in the atleast $k$-EXP ground truth datasets and GO term rarity on prediction accuracy.

## Results

### Topological similarity versus functional relatedness

Here, we provide evidence that the traditional assumption of NA methods, namely that topological similarity corresponds to functional relatedness, does not hold.

First, as a baseline, consider a network aligned to itself, i.e., to its 0% randomly perturbed counterpart. We know the correct node mapping, and pairs in this mapping can be considered functionally related. If we look at the topological similarity between pairs of nodes that should be aligned versus those that should not, we expect the former to be topologically identical, and the latter not to be. Also, we expect the distribution of topological similarities of the matching node pairs to be different than the distribution of topological similarities of non-matching pairs. Indeed, that is what we observe (Fig 2(a) and S2(a) Fig).

Now, consider a network aligned to its 25% randomly perturbed counterpart. Because only (a portion of) edges change, we still know the correct node mapping, i.e., which nodes are functionally related. At just 25% random perturbation (where networks are still 75% identical), we observe that the topological similarity distribution of node pairs that should be matched is now close to the topological similarity distribution of those that should not (Fig 2(b), and S2(b) and S3(b) Figs). In other words, the functionally matching nodes are only marginally more similar to each other than at random. So, even if the two networks being aligned are just a little different (and it is expected that PPI networks of different species are much more different than that), topological similarity is no longer correlated with functional relatedness. This fact holds for multiple prominent topological similarity measures, including GRAAL's [63] and MAGNA's [39] GDV similarity measure (Fig 2), GHOST's [13] spectral signature-based similarity measure (S2 Fig), and IsoRank's [47] PageRank-based similarity measure (S3 Fig). Note that while the three measures quantify topologically similarity in mathematically different ways, they all follow the general notion that a high score corresponds to neighborhood

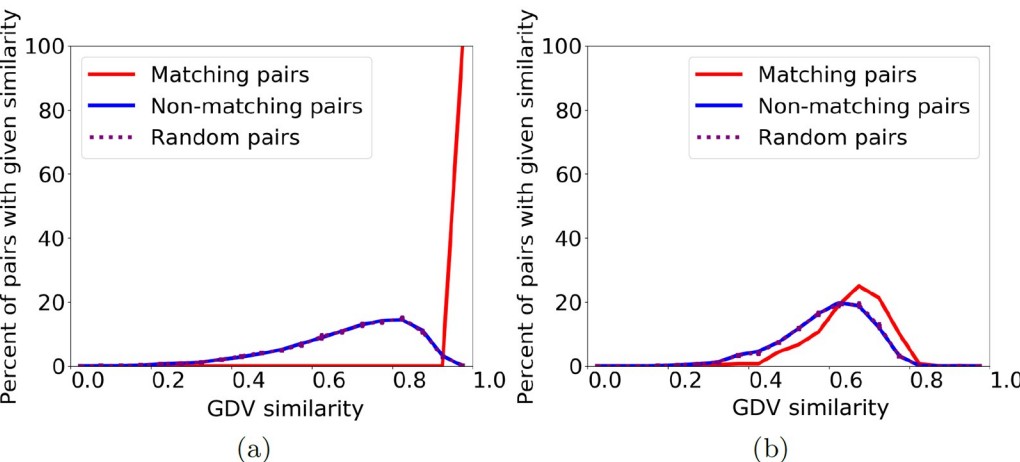

**Fig 2. Distribution of topological similarity (GDV similarity) between node pairs of a geometric random graph (i.e., a synthetic network) and its (a) 0% and (b) 25% randomly perturbed counterparts.** We show three lines representing the distribution of topological similarity for matching (i.e., functionally related) node pairs (blue), for non-matching, i.e., functionally unrelated, node pairs (red), and for 10 random samples of the same size as the set of matching pairs, averaged (purple). Results are qualitatively similar for 50% random perturbation, scale-free random graphs (a different type of synthetic networks), and GHOST's and IsoRank's similarity measures (S1–S3 Figs).

regions that are close to isomorphic (as discussed in Section "Introduction—Background and motivation"). Because all measures show qualitatively similar trends, and because GDV similarity was shown to outperform both GHOST's and IsoRank's similarities [14, 15], we focus on GDV similarity for the following analysis.

Second, we observe this trend, namely that the distributions of topological similarity for functionally related and functionally unrelated protein pairs are close to each other, for real world PPI networks as well (see below). Furthermore, we find that the distributions of sequence similarity are also close to each other for the two sets of protein pairs, and that distributions of the combination of topological and sequence similarities are close to each other as well. Specifically, we analyze the yeast and human PPI networks described in Section "Materials and methods—Data". Here, we consider proteins that share at least one experimentally inferred biological process GO term to be functionally related, proteins that do not share any GO terms to be functionally unrelated, proteins with GDV similarity of 0.85 or greater to be topologically similar [64], and proteins with E-value of $10^{-10}$ or lower to be sequence similar [65]. Our findings are as follows:

- Out of all functionally related protein pairs, only $\sim$28% are topologically similar (Fig 3(a), above the horizontal line), while even out of all functionally unrelated protein pairs, $\sim$14% are still topologically similar (Fig 3(b), above the horizontal line).

- Out of all functionally related protein pairs, $\sim$63% are sequence similar (Fig 3(a), to the right of the vertical line), while even out of all functionally unrelated protein pairs, $\sim$53% are still sequence similar (Fig 3(b), to the right of the vertical line).

- Out of all functionally related protein pairs, only $\sim$18% are both topologically and sequence similar (Fig 3(a), top right quadrant), while even out of all functionally unrelated protein pairs, only $\sim$8% are both topologically and sequence similar (Fig 3(b), top right quadrant).

In other words, functionally related nodes are only marginally more similar (for all types of similarity we consider) to each other than at random. Therefore, the existing NA assumption that is based on topological similarity fails, and instead our NA approach that is based on topological relatedness, TARA, is needed.

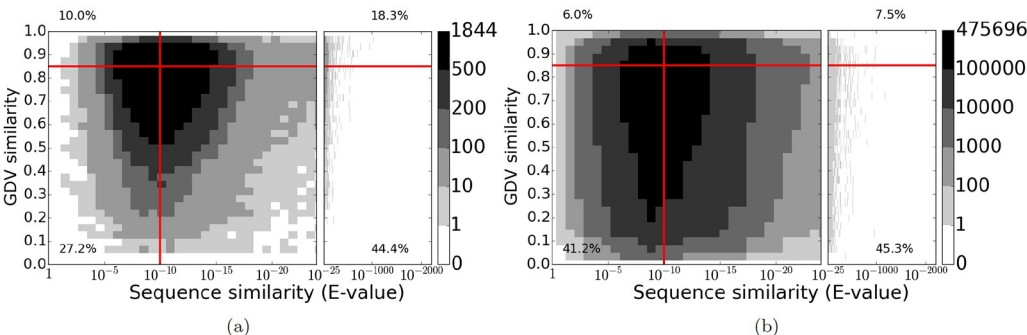

**Fig 3. Distribution of topological similarity (GDV similarity) versus sequence similarity (E-value) between yeast and human PPI networks of those yeast-human protein pairs that are (a) functionally related (i.e., share at least one biological process GO term such that the protein-GO term annotation was experimentally inferred) and (b) functionally unrelated (i.e., share zero GO terms).** The color of a pixel represents how many node pairs have a given topological similarity and given sequence similarity. The red horizontal and vertical lines indicate the thresholds for topologically similar ($y \geq 0.85$) or sequence similar ($x \leq 10^{-10}$) pairs, and the percentages indicate the fraction of pairs that are in a given quadrant.

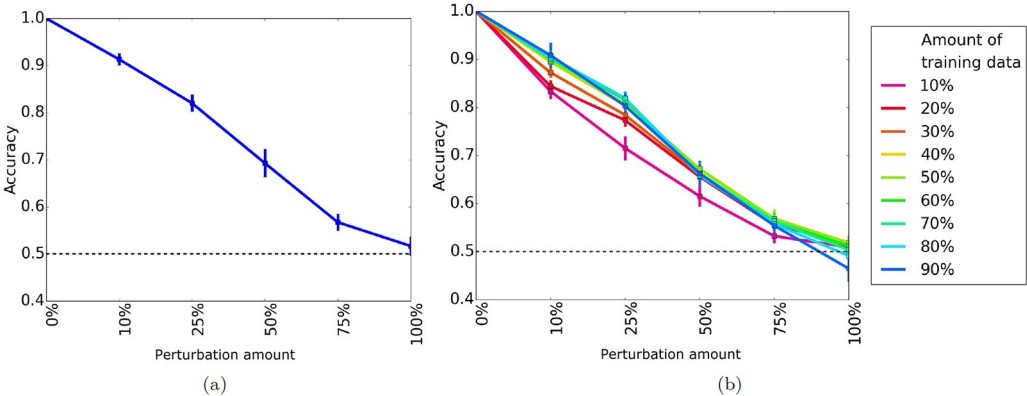

**Fig 4. Average prediction accuracy of (a) 10-fold cross-validation and (b) percent training tests for a geometric network and its randomly perturbed counterparts.** In panel (**b**), different colored lines represent how much data is used for training; these colors do not apply to panel (**a**). A dotted black line indicates the accuracy expected if the classifier makes random predictions. Qualitatively similar results for AUROC and for scale-free networks are shown in S4 and S5 Figs.

## TARA for network sets with known node mapping

**10-fold cross-validation.**   Here we evaluate TARA using 10-fold cross-validation. Specifically, for each network model (geometric and scale free), for each random perturbation level (0, 10, 25, 50, 75, 100), we obtain the average accuracy and average AUROC of the 10 folds. We expect that as the amount of random perturbation increases, prediction accuracy and AUROC decrease since the networks become more and more dissimilar. Indeed, this is what we observe (Fig 4(a) and S4 Fig). Also, we expect a random classifier to give around 50% accuracy since the class sizes are balanced; it will also have 50% AUROC by definition. This is empirically verified by the results at 100% random perturbation, where we are attempting to classify nodes between two completely different networks (Fig 4(a) and S4 Fig).

**Percent training tests.**   Again, we expect that as the amount of random perturbation increases, accuracy and AUROC decreases since networks are becoming more dissimilar. Also, we expect that as we increase the amount of training data, the accuracy and AUROC increases as well since more information is being used in the classifier. Overall, these are the trends we observe (Fig 4(b) and S5 Fig). We also see that using 90% of the data as training does not lead to drastic improvements; in fact, it is not always even the best. For some (geometric) networks, as low as 40% still gives comparable results. This is promising, as we do not necessarily have to rely on using a majority of the data for training.

These tests serve as a proof of concept that there is some learnable pattern between topological and functional relatedness, and so it makes sense to continue our study for real-world networks.

## TARA for network sets with unknown node mapping

**10-fold cross-validation.**   Again, we evaluate TARA using 10-fold cross-validation. We expect that as $k$ increases, accuracy and AUROC do as well since the conditions for a pair of proteins to be functionally related becomes more stringent. Indeed, this is what we observe (Fig 5(a), S6 Fig).

**Percent training tests.**   We see similar results for percent training as we do for 10-fold cross-validation (Fig 5(b), S7 Fig). Note that unlike percent training for synthetic networks, the amount of training data has very little effect on accuracy except for atleast3-EXP. This may

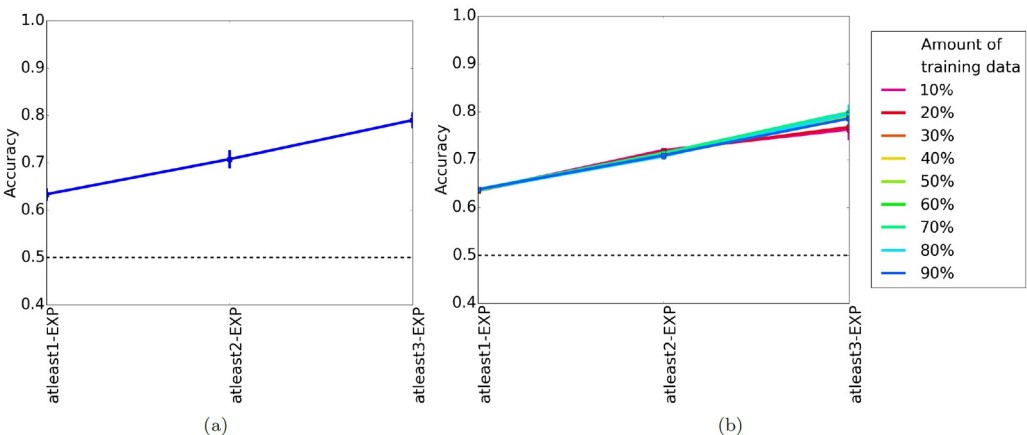

**Fig 5. Average prediction accuracy of (a) 10-fold cross-validation and (b) percent training tests for real-world networks.** In panel (b), different colored lines represent how much data is used for training; these colors do not apply to panel (a). A dotted black line indicates the accuracy expected if the classifier makes random predictions. Qualitatively similar results for AUROC are shown in S6 and S7 Figs.

be because atleast1-EXP and atleast2-EXP contain a lot more data, meaning even a small percentage is enough to train a good classifier.

Overall, we are able to detect a pattern between topological relatedness and functional relatedness. So, it makes sense to generate an alignment and evaluate TARA in the protein function prediction task.

## TARA for protein function prediction

Here, we evaluate TARA and existing NA methods in the task of protein function prediction. Specifically, we take the alignments generated from each method and put them through the protein function prediction framework as described above. We first compare TARA's percent training tests to each other, and then we compare TARA to existing NA methods.

**Comparing TARA's percent training tests to each other.** For simplicity, we only compare a subset of TARA's percent training tests. Specifically, because classification accuracy does not vary significantly between different percent training tests, we focus on the extremes (10 and 90) and the middle (50). So, we have 27 total evaluation tests for TARA, resulting from combinations of the three percent training tests and the nine ground truth-rarity datasets discussed above.

We expect that as we increase the amount of training data (10 to 50 to 90), precision will increase and recall will decrease. This is because more training data means the classifier will likely be better (increasing precision), but will result in less testing data and thus smaller alignments and fewer predictions (decreasing recall). Similarly, we expect that as we increase $k$ in our atleast $k$-EXP ground truth datasets, precision will increase and recall will decrease. This is because at higher $k$, we will be training on higher quality data (increasing precision), but there will be less data overall, resulting in smaller alignments and fewer predictions (decreasing recall). Finally, we expect that as we consider rarer GO terms, precision will increase and recall will decrease. Intuition from existing studies suggests that rarer GO terms are more meaningful [62], so the data will be higher quality (increasing precision), but again there will be less data overall (decreasing recall). Indeed, we observe these trends (Fig 6 and S8 Fig) for all but atleast3-EXP at the 50 and 25 rarity thresholds; there is not enough data for TARA to generate

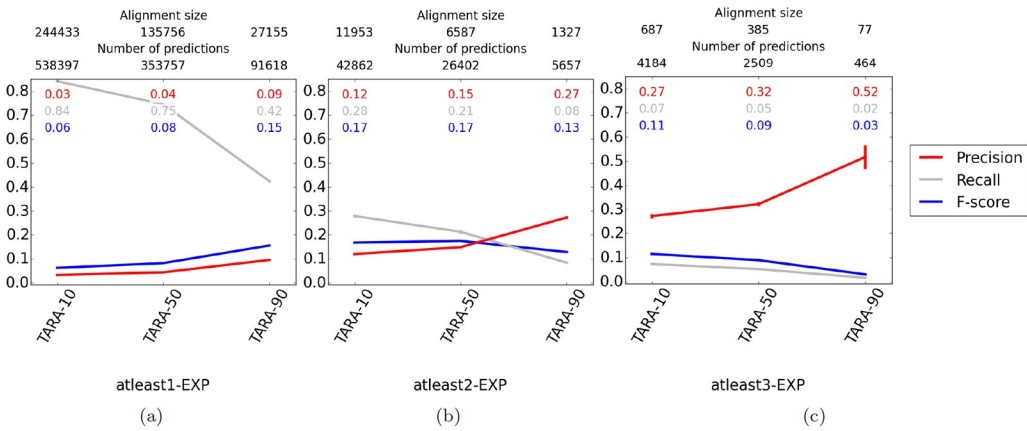

**Fig 6. Comparison of different TARA evaluation tests in the task of protein function prediction, for the ALL GO term rarity threshold.** Different percent training tests, specifically 10, 50, and 90, are compared within each panel, and different ground truth datasets, specifically **(a)** atleast1-EXP, **(b)** atleast2-EXP, and **(c)** atleast3-EXP, are compared across panels. The alignment size (i.e., the number of aligned yeast-protein pairs) and number of functional predictions (i.e., predicted protein-GO term associations) made by each method, averaged over the 10 instances we perform for each test, are shown on the top. For example, the alignment for TARA-90 for the atleast2-EXP dataset contains 1,327 aligned yeast-human protein pairs, and predicts 5,657 protein-GO term associations. Raw precision, recall, and F-score values are color-coded inside each panel. Complete results for the other rarity thresholds are shown in S8 Fig.

alignments or make predictions for those parameters. Inability to learn on small datasets is one drawback of machine learning methods in general, not just TARA.

In order to simplify comparisons between TARA and existing NA methods, we choose a representative percent training test (i.e., either TARA-10, TARA-50, or TARA-90) for each of the nine ground truth-rarity datasets discussed previously. In other words, we go from 27 TARA evaluation tests to nine (though we actually have seven since TARA does not make predictions for atleast3-EXP at the 50 and 25 rarity thresholds, per the above discussion). Generally, we try to choose the percent training test that has both high precision (meaning predictions are accurate) and a large number of predictions (meaning we uncover as much biological knowledge as possible), as these represent TARA's best results. The choices are given in Table 2.

**Comparing TARA against existing NA methods.** We compare against three existing methods, WAVE, SANA, and PrimAlign. We compare against these three methods for the following reasons (also, see Section "Introduction"). WAVE and SANA are state-of-the-art methods that use graphlets, just like TARA, allowing us to fairly analyze how much TARA's supervised process helps. Also, they operate under the assumption that topologically similar nodes are functionally related, which is what TARA challenges. However, recall that WAVE and SANA are one-to-one methods, while TARA is a many-to-many method. So, we analyze PrimAlign, because it is a state-of-the-art many-to-many method. In addition, PrimAlign operates under the assumption that we challenge, namely that topologically similar nodes are functionally related. Recall from Section "Introduction—Related work" that WAVE, SANA,

**Table 2. Representative choices of TARA's percent training tests for each of the 9 ground truth datasets.**

|  | ALL | 50 | 25 |
|---|---|---|---|
| atleast1-EXP | TARA-90 | TARA-90 | TARA-90 |
| atleast2-EXP | TARA-90 | TARA-10 | TARA-10 |
| atleast3-EXP | TARA-90 | N/A | N/A |

and PrimAlign were already shown to outperform a number of previous NA methods, and hence, we believe that comparing to these three methods is sufficient. Also, keep in mind that a theoretical precision of one is not possible with TARA, unlike WAVE, SANA, and PrimAlign. This is because TARA uses part of the ground truth data for training, meaning it impossible to make predictions for that portion. In other words, TARA is inherently disadvantaged compared to existing methods.

In more detail, WAVE and SANA use graphlet-based topological information like TARA (however, keep in mind that sequence information or any other data could also be used in TARA, which is subject of our future work). Specifically, WAVE and SANA both use GDV similarity to score the similarities of node pairs, and SANA also uses an equal weighing of node conservation and edge conservation (i.e., we set both `s3` and `esim` to 1). Unlike WAVE and SANA, PrimAlign uses both topological (PageRank-based) information and sequence similarity (negative log of E-value) information by default. Specifically, regarding the latter PrimAlign study, which analyzes the same yeast and human PPI networks as we do, considers all sequence similar proteins between the networks with an E-value $\leq 10^{-7}$, which results in 55,594 sequence similarity-based anchor links. We run this default version, called PrimAlign-TS. We also analyze a topological version of PrimAlign (PrimAlign-T) for fair comparison with TARA, which in this study uses topological but not sequence information. To create an as fairly comparable as possible topological version of PrimAlign, we instead use the 55,594 most topologically (GDV) similar yeast-human protein pairs as anchor links. Lastly, we are also interested in using sequence information only (Sequence, or S), in order to better understand the effect of T or S alone. We do so by taking those 55,594 sequence similar pairs from PrimAlign-TS and treating them as the alignment, disregarding any topological information from the PPI networks.

Summarizing the different NA methods, TARA, WAVE, SANA, and PrimAlign-T use topological information, Sequence uses sequence information, and PrimAlign-TS uses both topological and sequence information. Furthermore, recall that the different methods have different levels of comparability to TARA in terms of information used (T versus S versus TS) and alignment type (one-to-one versus many-to-many) (Table 3). To show that our assumption holds, namely that topologically related, rather than topologically similar, nodes should be aligned, it would be sufficient to show that TARA, a T method, outperforms the other T methods. If TARA, a T method, also outperforms Sequence or PrimAlign-TS, then this would further underscore the need of a data-driven approach like ours.

We discuss our results below (Fig 7 and S9 Fig). Note that we primarily focus on precision because in terms of potential wet lab validation of some predictions, we believe it is more important to have fewer but mostly correct predictions (e.g., 90 correct out of 100 made) than a greater number of mostly incorrect predictions (e.g., 300 correct out of 1000 made). While in the latter example more predictions are correct (300 versus 90), leading to a higher (almost

**Table 3. Comparability of the existing methods considered in this study to TARA in terms of type of information used (T versus S versus TS) and alignment type (one-to-one versus many-to-many).**

| Existing NA method | Fair to TARA in terms of: | |
|---|---|---|
| | Information used | Alignment type |
| WAVE | Yes | No |
| SANA | Yes | No |
| Sequence | No | Yes |
| PrimAlign-T | Yes | Yes |
| PrimAlign-TS | No | Yes |

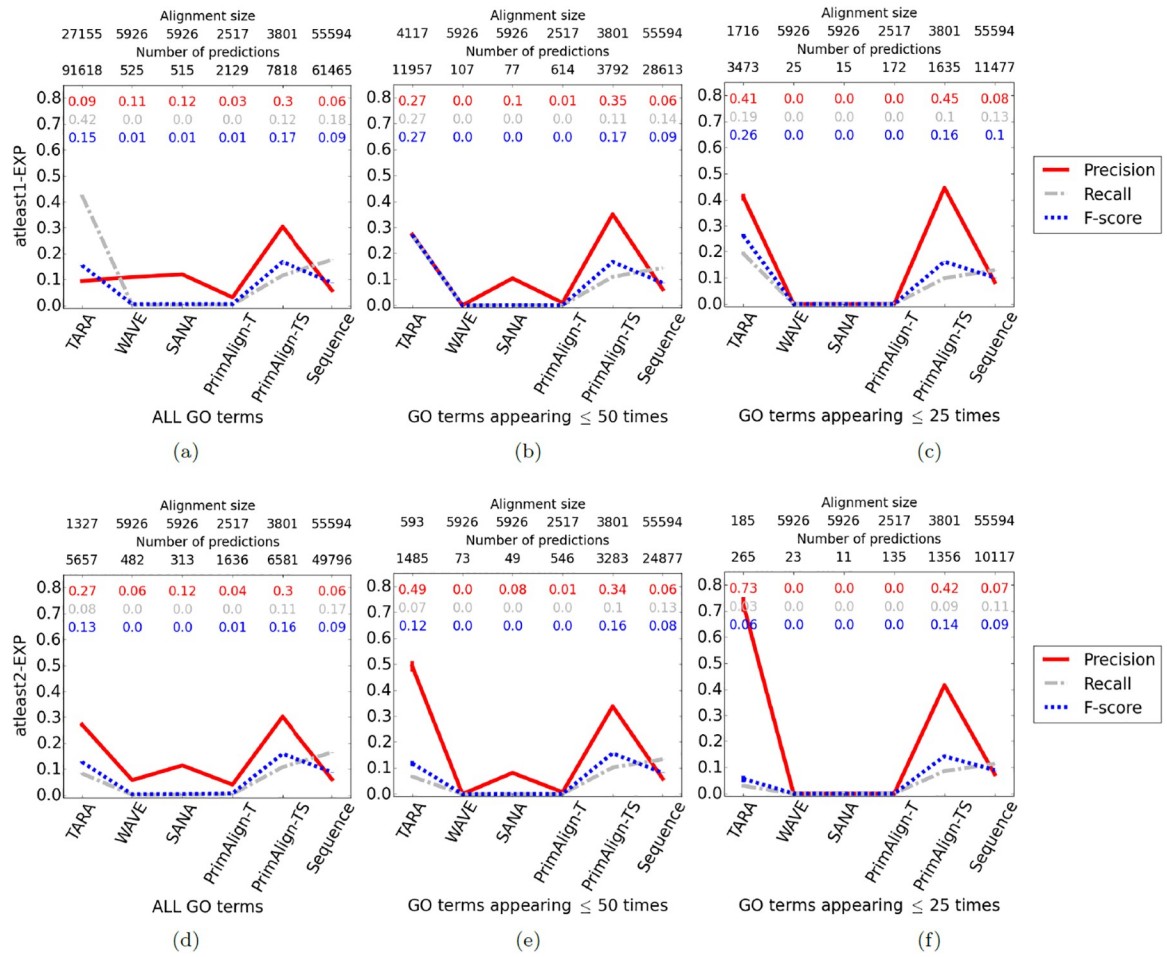

**Fig 7. Comparison of the six considered NA methods for rarity thresholds (a, d) ALL, (b, e) 50, and (c, f) 25 using ground truth datasets (a, b, c) atleast1-EXP and (d, e, f) atleast2-EXP in the task of protein function prediction.** The alignment size (i.e., the number of aligned yeast-protein pairs) and number of functional predictions (i.e., predicted protein-GO term associations) made by each method. For example, the alignment for TARA in panel **(a)** contains 27,155 aligned yeast-human protein pairs, and predicts 91,618 protein-GO term associations. Raw precision, recall, and F-score values are color-coded inside each panel. Results for atleast3-EXP are shown in S9 Fig.

triple) recall, many more are also incorrect, leading to lower precision (0.3 versus 0.9). However, we do not completely discount recall and F-score, as they may still be valuable measures for other considerations. Also, keep in mind that the expected precision and recall for a random alignment is near 0. A random alignment is not expected to match functionally related proteins, meaning essentially random protein-GO term associations will be predicted.

- Compared to other T methods, TARA is superior to WAVE and SANA in 6/7 tests with respect to precision, and in all seven tests with respect to recall (the seven tests are summarized in Table 2). Importantly TARA is always superior to PrimAlign-T, the most fairly comparable method to TARA, in terms of both precision and recall. These trends support our claim that topologically related, not topologically similar, nodes are the ones that are functionally related.

- Compared to PrimAlign-TS, TARA is superior in 3/7 tests (atleast2-EXP for the 50 and 25 rarity thresholds, and atleast3-EXP for ALL GO terms) with respect to precision. Of the

remaining four tests, TARA is superior in two and comparable in two with respect to F-score.

- Compared to Sequence, TARA is superior in all seven tests in terms of precision, and superior in 3/7 in terms of recall. Of those remaining four tests, it is still superior in two of them with respect to F-score.

An interesting note is that the precision of PrimAlign-TS is much greater than simply the sum of precision from Sequence and PrimAlign-T, suggesting that combining topological and sequence information in a meaningful way can have compounded effects. This is promising for incorporating sequence information into TARA, which is our future work.

While precision, recall, and F-score are important overall measures, it is also necessary to zoom into the actual predictions that the methods make. We focus on TARA and PrimAlign-TS, as these two methods perform the best, with the parameters from Fig 7.

We see that for atleast1-EXP, no matter the rarity threshold, TARA makes many more predictions than PrimAlign-TS, and yet still has comparable precision for the 50 and 25 GO term rarity thresholds (Fig 7). In other words, TARA is potentially uncovering more biological knowledge than PrimAlign-TS but with similar accuracy. For atleast2-EXP, for the ALL GO term rarity threshold, TARA and PrimAlign-TS make a similar number of predictions with similar precision, and for the 50 and 25 rarity thresholds, TARA outperforms PrimAlign-TS, though at the cost of fewer predictions (Fig 7). For atleast3-EXP, for the ALL GO term rarity threshold, TARA outperforms PrimAlign-TS, also at the cost of fewer predictions (S9 Fig).

Importantly, we see that the number of predictions in the overlap of TARA and PrimAlign-TS is generally small (Fig 8 and S10 Fig), suggesting that most of the two methods' predictions are complementary. Therefore, we can say that TARA has some advantage in every case (whether it be precision, recall, or number of predictions), and at worst complements PrimAlign, which even uses sequence information that TARA does not. This, in addition to TARA outperforming WAVE and SANA, justifies the need for introducing our new data-driven approach.

We also look at the time it takes to obtain an alignment for TARA, WAVE, SANA, PrimAlign-T, and PrimAlign-TS for the ALL GO term rarity threshold, as the given threshold has the most data and thus will be the worst case time-wise out of all thresholds. We do not consider Sequence as we did not compute any alignment in this case; instead, the alignment was included from the PrimAlign study. We expect as that $k$ (in the atleast $k$-EXP ground truth dataset) increases, the time for TARA to produce an alignment decreases since there is less (but higher quality) data overall, and thus less data to train on. This is what we observe (Table 4). Regarding the existing NA methods, WAVE uses a seed-and-extend alignment strategy, which is expected to take some time. The running time of SANA is a parameter, which we choose to be 60 minutes since SANA requires such time to find a good alignment for networks of the sizes we analyze. We find that WAVE and SANA are both slower than TARA for atleast2-EXP and atleast3-EXP, and SANA is comparable to TARA for atleast1-EXP, meaning that TARA is overall both faster and more accurate at predicting protein function than the two one-to-one NA methods. Lastly, we find that PrimAlign and its variants are fast, which is expected because the method is linear in the number of edges.

## A closer look at TARA

We also explore why TARA is able to outperform the traditional NA methods. Recall the distributions of topological similarity scores (which traditional NA methods use) from Section

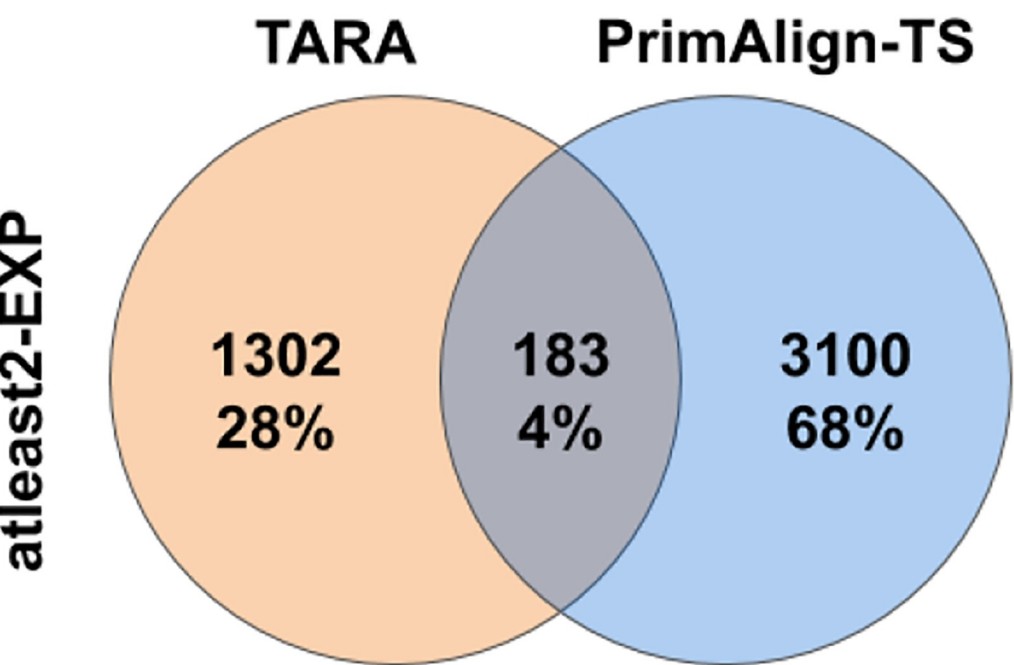

**Fig 8. Overlap of the functional predictions made by TARA and PrimAlign for atleast2-EXP at the 50 rarity threshold.** Percentages are out of the total number of unique predictions made by both methods combined. Complete results for all methods and parameters are shown in S10 Fig and S1 Table.

"Results—Topological similarity versus functional relatedness". When the two networks are just a bit different from each other, nodes that should be matched (i.e., are functionally related) are only marginally more topologically similar to each other than at random, leading to suboptimal alignments. If we analyze TARA's topological relatedness scores (described below) in the same way, we find that TARA can better distinguish matching node pairs from non-matching node pairs. This could explain why TARA outperforms the traditional NA methods.

**Table 4. Running time (rounded to the nearest second) comparison of TARA, WAVE, SANA, PrimAlign-T, and PrimAlign-TS for ALL GO terms.**

| Running time (s) | atleast1-EXP | atleast2-EXP | atleast3-EXP |
|---|---|---|---|
| TARA | 3,642 | 210 | 168 |
| WAVE | 1,686 | 1,686 | 1,686 |
| SANA | 3,600 | 3,600 | 3,600 |
| Sequence | N/A | N/A | N/A |
| PrimAlign-T | 3 | 3 | 3 |
| PrimAlign-TS | 16 | 16 | 16 |

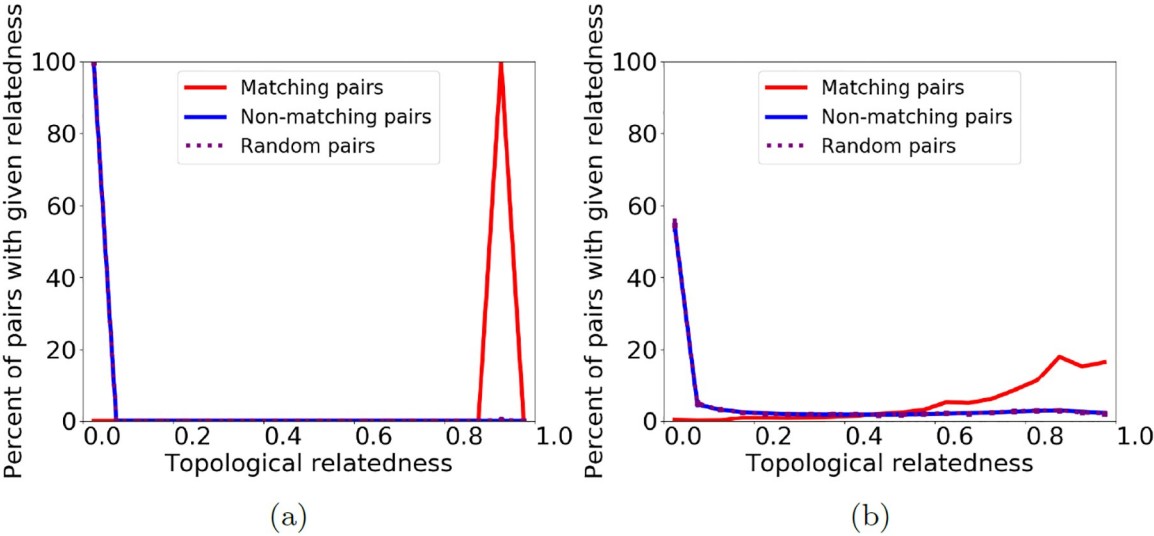

**Fig 9. Distribution of TARA's redefined topological relatedness between node pairs of a geometric random graph (i.e., a synthetic network) and its (a) 0% and (b) 25% randomly perturbed counterparts.** We show three lines representing the distribution of topological relatedness for matching (i.e., functionally related) node pairs (blue), for non-matching, i.e., functionally unrelated node pairs (red), and for 10 random samples of the same size as the set of matching pairs, averaged (purple).

To extract topological relatedness scores from TARA's framework, we do the following. Consider the 90% training test (while this applies to any percent training test, we focus on 90 because TARA-90 generally performs the best), where we first train a classifier on 90% of a balanced dataset. Then, instead of evaluating on the remaining 10% of the data as above, we input the feature vector of each node pair across networks into the trained classifier. Rather than directly outputting whether a pair is functionally related or not, we obtain the *probability* that the two nodes are functionally related instead. We can interpret this probability as a redefined "relatedness" measure, where now nodes are topologically related if they are likely to be functionally related.

Then, mirroring our initial analyses (Fig 2), we examine the distributions of these topological relatedness scores on the same networks and random perturbation levels. For a geometric network and its 0% randomly perturbed counterpart, we again see a distinct difference between the distributions of matching pairs and all pairs (Fig 9(a)). But, even for 25% random perturbation, the distributions are now different from each other (Fig 9(b)), and this difference is greater than the difference in distributions of the equivalent topological (GDV) similarity scores (Fig 2(b)). In other words, TARA's topological relatedness scores are better able to distinguish matching node pairs from non-matching node pairs compared to traditional topological similarity scores, which could explain the superior results of TARA over traditional NA methods. Improving these learned topological relatedness scores (e.g., so that the difference in distributions at 25% random perturbation looks like the difference at 0% random perturbation), and using them to produce alignments that are more fairly comparable to some traditional NA methods (e.g., to produce one-to-one alignments) are subjects of our future work.

## Generalizability of TARA

Just like with any supervised classification approach, the key goal of TARA is to first train the approach on the training portion of the compared networks, and then to test it on the testing portion of the same networks that was hidden during the training process, in order to validate

that the approach is accurate on the *known* but hidden knowledge from the testing data. Then, the goal is to retrain TARA on *all* of the (training plus testing) data in order to predict *novel* knowledge from the same data.

Just like any supervised classification approach in the context of any problem, for highest accuracy in the context of the NA problem, TARA should ideally be trained for each new pair of networks considered. That is, TARA when trained on one pair of networks (using only the training portion of the data), is expected to be at least as accurate when tested on the same pair of networks (using only the testing portion of the data) than when tested on a different pair of networks.

Of course, training TARA on a new pair of networks of interest is possible only if there exists data on whether a node pair is functionally related or not, for those networks. If such data does not exist, i.e., if one cannot determine for the new networks of interest whether a node pair is functionally related or not, then one must apply a TARA instance pre-trained on a different pair of networks to the new pair of networks.

So, here, we investigate how well TARA performs in this task, i.e., how generalizable it is. Namely, we examine whether we can train TARA on one pair of networks and apply the trained TARA instance to a new pair of networks to still accurately predict functional knowledge. Specifically, we apply TARA, trained on the yeast and human PPI networks as described in Section "Materials and methods—Data" ("2017 networks"), to more recent yeast and human PPI networks from the same database ("2020 networks"). The new networks come from BioGRID version 3.5.181 accessed in February 2020; like for the 2017 networks, we again only include physical interactions.

Details of our experimental setup for this analysis are as follows.

- We repeat the exact same *training* process as before, i.e., on the 2017 networks. Namely, on these networks, we create the same 10 balanced datasets, and, for a given balanced dataset, for each ground truth-rarity dataset and $y$% training amount, we split the data into $y$% training and $(100 - y)$% testing. Then, we train a logistic regression classifier using the GDVdiff feature for a node pair based on the 2017 networks. So, after these steps, we have trained TARA on the same node pairs and features vectors as before.

- But then, we perform *testing* on the 2020 networks. That is, of the node pairs in the 2017-network-based testing set from the previous bullet, we keep only those pairs in which both nodes are present in both the 2017 and 2020 network data. For the resulting node pairs, we compute their new node pair feature vectors based on the 2020 (rather than 2017) networks. We feed the new 2020-network-based feature vectors into the 2017-network-trained classifier, and add any node pair predicted as functionally related to the 2020-network-based alignment. Finally, this alignment is used in the protein function prediction framework. In this way, we can fairly compare results between the 2017-network-based alignments (computed in previous sections) and the corresponding 2020-network-based alignments (computed as just described), since all training is done on the same node pairs with the same 2017-network-based feature vectors, and the testing only differs in which network set (2017 versus 2020) the feature vectors were extracted from. Algorithm 4 outlines this process.

- Repeating for each balanced dataset, we obtain 10 precision, recall, and F-score values, and we average over the 10 values for each measure.

**Algorithm 4** Applying a pre-trained TARA instance to a new pair of networks. Given two networks $G_1$ and $G_2$, the set of conditions $R$ that a node pair needs to satisfy to be in a given ground truth dataset, the set of conditions $R'$ that a node pair needs to satisfy to be considered

not functionally related, random seed state *d*, a *y* percent training amount, and two networks $G_3$ and $G_4$, return an alignment of $G_3$ and $G_4$. For notations and their meanings, see Table 1

```
1: Initialize set A to be empty.
2: let S_p, S_n = bal(G_1, G_2, R, R', d), as computed by Algorithm 2.
3: let Tr_p = random.sample(S_p, ⌊y|S_p|/100⌋, d).
4: let Te_p = S_p\Tr_p
5: let Tr_n = random.sample(S_n, ⌊y|S_n|/100⌋, d).
6: let Te_n = S_n\Tr_n.
7: let Tr = Tr_p ∪ Tr_n.
8: let Te = Te_p ∪ Te_n.
9: Train a predictive function, LogReg : ℝ^73 → {0,1}, with logistic
regression, on Tr, where the feature vector of node pair s_ij ∈ Tr is
given by g(s_ij, G_1, G_2), and the label of s_ij is {1 if s_ij ∈ S_p, 0 if
s_ij ∈ S_n}.
10: Lines 1-9 are identical to those of Algorithm 3, representing the
fact that the training processes are identical (assuming that G_1, G_2,
R, R', d, and y do not change between Algorithms 3 and 4).
11: for each s_ij ∈ Te do
12:   if s_ij ∈ V_3 × V_4 do
13:     let x_ij = g(s_ij, G_3, G_4)
14:     if LogReg(x_ij) = 1 then
15:       A.add(s_ij)
16:     end if
17:   end if
18: end for
19: return A
For example, if G_1 is the 2017 yeast PPI network, G_2 is the 2017 human
PPI network, R is the set of conditions for the atleast3-EXP ground
truth dataset, R' is the set of conditions for a protein pair to be
considered not functionally related (i.e., shared no GO terms of any
kind), d is 0, y is 90%, G_3 is the 2020 yeast PPI network, and G_4 is
the 2020 human PPI network, getAln*(G_1, G_2, R, R', d, y, G_3, G_4) returns
the alignment between the 2020 yeast and human PPI networks generated
by a classifier trained on functionally related protein pairs defined
by R, functionally unrelated pairs defined by R', using a random seed
state of 0 for sampling, and 90% of the 2017-network-based data for
training.
```

Our findings are as follows. By looking at the number of protein function predictions, TARA applied to the 2020 networks generally results in somewhat fewer predictions compared to TARA applied to the 2017 networks. Even if the two TARA versions are equally accurate, the differing number of predictions alone could naturally result in the former having somewhat higher precision and somewhat lower recall. Indeed, this is exactly what we observe (Fig 10). A key result is that the two TARA versions are quite comparable, i.e., that TARA trained on the 2017 networks, when it is tested on the 2020 networks, results in pretty similar (somewhat higher) precision and (somewhat lower) recall values as when it is tested on the 2017 networks. This is extremely encouraging, as it indicates that TARA *is* generalizable in our considered test.

## Conclusion

We present TARA as a method that challenges the assumption of current NA methods that topologically similar nodes are functionally related. We have shown that given the topological feature vector of a pair of nodes, TARA can accurately predict whether the nodes are functionally related. In other words, we have designed a method that can detect from training data a pattern between topological relatedness and functional relatedness in both synthetic and

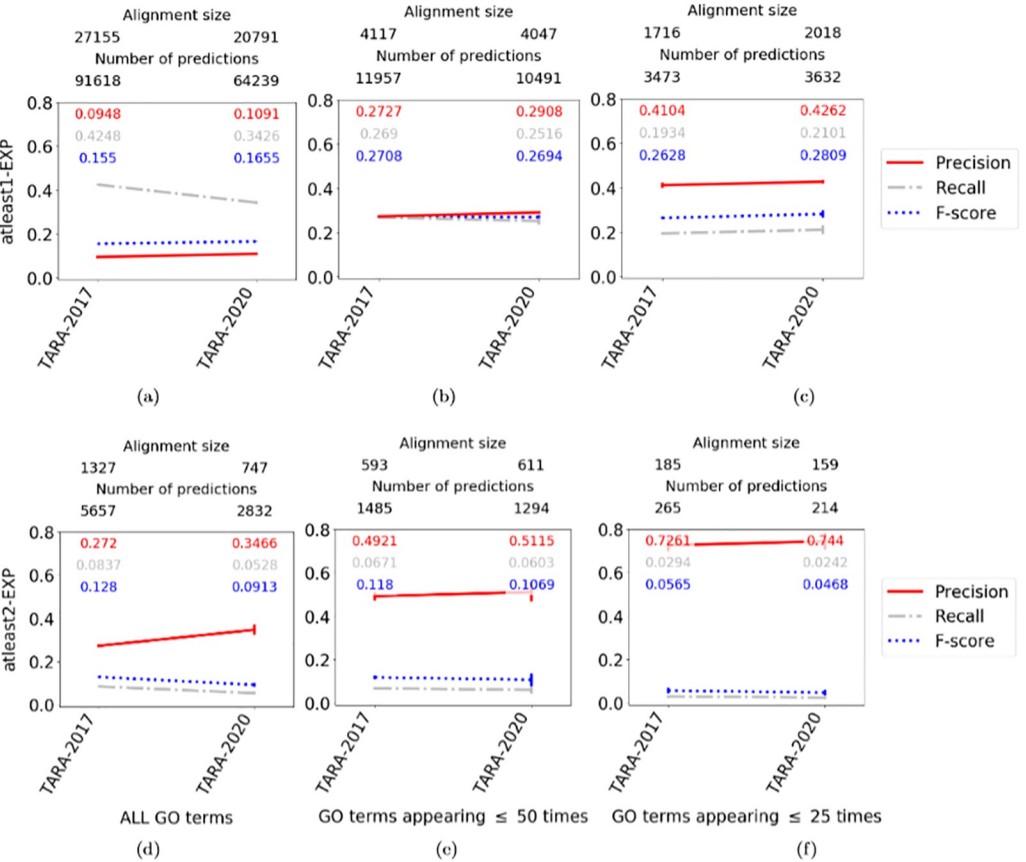

**Fig 10. Comparison of TARA on the 2017 versus 2020 networks for rarity thresholds (a, d) ALL, (b, e) 50, and (c, f) 25 using ground truth datasets (a, b, c) atleast1-EXP and (d, e, f) atleast2-EXP in the task of protein function prediction.** The alignment size (i.e., the number of aligned yeast-protein pairs) and number of functional predictions (i.e., predicted protein-GO term associations) made by each method. For example, the alignment for TARA-2017 in panel **(a)** contains 27,155 aligned yeast-human protein pairs, and predicts 91,618 protein-GO term associations. Raw precision, recall, and F-score values are color-coded inside each panel. Results for atleast3-EXP are shown in S11 Fig.

real-world networks. Then, taking pairs predicted as functionally related from the testing data as an alignment, we have shown that TARA generally outperforms or complements existing approaches, even those that use sequence similarity-based anchor links across network as input (unlike TARA), in the task of protein function prediction, one of the ultimate goals of NA. As such, TARA provides researchers with a valuable data-driven approach to NA and protein function prediction.

To our knowledge, TARA is the first data-driven NA approach. As such, it is just a proof-of-concept. There are many directions in which this work can be taken. For one, we use a relatively simple GDV-based feature of a node pair. However, more sophisticated combinations of GDVs could be explored. Other embedding methods (i.e., ways to extract feature vectors of nodes) such as matrix factorization [19] or graph convolution networks [52] could show improvement. Also, including sequence similarity-based anchor links like PrimAlign does, is promising, especially given the fact that combining topological and sequence information seems to have compounding effects. Also, we train a simple classifier—logistic regression—but potential improvement could be seen with more sophisticated models. Furthermore, in this study we have focused on pairwise, homogeneous, and static NA. However, there has been

work in aligning multiple [21, 22, 66, 67], heterogeneous [68, 69], or dynamic [70–72] networks. Our general framework could be adapted to each of these types of NA.

## Supporting information

**S1 Fig. Distribution of GDV similarity.** Distribution of topological similarity (GDV similarity) between node pairs of a **(a,b,c)** geometric and **(d,e,f)** scale-free network and their **(a,d)** 0%, **(b,e)** 25% randomly perturbed, and **(c,f)** 50% randomly perturbed counterparts. We show three lines representing the distribution of topological similarity for matching, i.e., functionally related, node pairs (blue), for non-matching, i.e., functionally unrelated, node pairs (red), and for 10 random samples of the same size as the set of matching pairs, averaged (purple). (PDF)

**S2 Fig. Distribution of GHOST's topological similarity measure.** Distribution of topological similarity (GHOST) between node pairs of a **(a,b,c)** geometric and **(d,e,f)** scale-free network and their **(a,d)** 0%, **(b,e)** 25% randomly perturbed, and **(c,f)** 50% randomly perturbed counterparts. We show three lines representing the distribution of topological similarity for matching, i.e., functionally related, node pairs (blue), for non-matching, i.e., functionally unrelated, node pairs (red), and for 10 random samples of the same size as the set of matching pairs, averaged (purple). (PDF)

**S3 Fig. Distribution of IsoRank's topological similarity measure.** Distribution of topological similarity (IsoRank) between node pairs of a **(a,b,c)** geometric and **(d,e,f)** scale-free network and their **(a,d)** 0%, **(b,e)** 25% randomly perturbed, and **(c,f)** 50% randomly perturbed counterparts. We show three lines representing the distribution of topological similarity for matching, i.e., functionally related, node pairs (blue), for non-matching, i.e., functionally unrelated, node pairs (red), and for 10 random samples of the same size as the set of matching pairs, averaged (purple). (PDF)

**S4 Fig. Accuarcy and AUROC of 10-fold cross-validation for geometric and scale-free networks.** Average **(a,b)** prediction accuracy and **(c,d)** AUROC of 10-fold cross-validation for **(a, c)** geometric and **(b,d)** scale-free networks. (PDF)

**S5 Fig. Accuracy and AUROC of percent training tests for geometric and scale-free networks.** Average **(a,b)** prediction accuracy and **(c,d)** AUROC of percent training tests for **(a,c)** geometric and **(b,d)** scale-free networks. (PDF)

**S6 Fig. Accuarcy and AUROC of 10-fold cross-validation for real-world networks.** Average **(a)** prediction accuracy and **(b)** AUROC of 10-fold cross-validation for real-world networks. (PDF)

**S7 Fig. Accuracy and AUROC of percent training tests for real-world networks.** Average **(a)** prediction accuracy and **(b)** AUROC of percent training tests for real-world networks. (PDF)

**S8 Fig. Comparison of different TARA evaluation tests.** Comparison of different TARA evaluation tests in the task of protein function prediction, for GO term rarity thresholds **(a, d, g)** ALL, **(b, e)** 50, and **(c, f)** 25 using ground truth datasets **(a, b, c)** atleast1-EXP, **(d, e, f)** atleast2-EXP, and **(g)** atleast3-EXP. Different percent training tests, specifically 10, 50, and 90,

are compared within each panel. The alignment size (i.e., the number of aligned yeast-protein pairs) and number of functional predictions (i.e., predicted protein-GO term associations) made by each method, averaged over the 10 instances we perform for each test, are shown on the top. For example, the alignment for TARA-90 in panel **(a)** contains 27,155 aligned yeast-human protein pairs, and predicts 91,618 protein-GO term associations. Raw precision, recall, and F-score values are color-coded inside each panel.
(PDF)

**S9 Fig. Comparison of the six considered NA methods.** Comparison of the six considered NA methods for rarity thresholds **(a, d, g)** ALL, **(b, e)** 50, and **(c, f)** 25 using ground truth datasets **(a, b, c)** atleast1-EXP, **(d, e, f)** atleast2-EXP, and **(g)** atleast3-EXP in the task of protein function prediction. The alignment size (i.e., the number of aligned yeast-protein pairs) and number of functional predictions (i.e., predicted protein-GO term associations) made by each method. For example, the alignment for TARA in panel **(a)** contains 27,155 aligned yeast-human protein pairs, and predicts 91,618 protein-GO term associations. Raw precision, recall, and F-score values are color-coded inside each panel.
(PDF)

**S10 Fig. Overlap of the functional predictions made by TARA and PrimAlign.** Overlap of the functional predictions made by TARA and PrimAlign for GO term rarity thresholds **(a, d, g)** ALL, **(b, e)** 50, and **(c, f)** 25 using ground truth datasets **(a, b, c)** atleast1-EXP, **(d, e, f)** atleast2-EXP, and **(g)** atleast3-EXP. Percentages are out of the total number of unique predictions made by both methods combined.
(PDF)

**S11 Fig. Comparison of TARA on the 2017 versus 2020 networks.** Comparison of TARA on the 2017 versus 2020 networks for rarity thresholds **(a, d, g)** ALL, **(b, e)** 50, and **(c, f)** 25 using ground truth datasets **(a, b, c)** atleast1-EXP, **(d, e, f)** atleast2-EXP, and **(g)** atleast3-EXP in the task of protein function prediction. The alignment size (i.e., the number of aligned yeast-protein pairs) and number of functional predictions (i.e., predicted protein-GO term associations) made by each method. For example, the alignment for TARA-2017 in panel **(a)** contains 27,155 aligned yeast-human protein pairs, and predicts 91,618 protein-GO term associations. Raw precision, recall, and F-score values are color-coded inside each panel.
(PDF)

**S1 Table. Detailed statistics regarding predictions made by TARA and PrimAlign-TS.**
(PDF)

## Author Contributions

**Conceptualization:** Shawn Gu, Tijana Milenković.

**Formal analysis:** Shawn Gu.

**Funding acquisition:** Tijana Milenković.

**Investigation:** Shawn Gu.

**Methodology:** Shawn Gu, Tijana Milenković.

**Software:** Shawn Gu.

**Supervision:** Tijana Milenković.

**Visualization:** Shawn Gu.

**Writing – original draft:** Shawn Gu.

**Writing – review & editing:** Shawn Gu, Tijana Milenković.

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
