## [Decision Letter · Decision Letter 0]

11 Feb 2020

PONE-D-19-28710

Data-driven network alignment

PLOS ONE

Dear Mr. Gu,

Thank you for submitting your manuscript to PLOS ONE. After careful consideration, we feel that it has merit but does not fully meet PLOS ONE’s publication criteria as it currently stands. Therefore, we invite you to submit a revised version of the manuscript that addresses the points raised during the review process.

We would appreciate receiving your revised manuscript by Mar 27 2020 11:59PM. To enhance the reproducibility of your results, we recommend that if applicable you deposit your laboratory protocols in protocols.io, where a protocol can be assigned its own identifier (DOI) such that it can be cited independently in the future. For instructions see: http://journals.plos.org/plosone/s/submission-guidelines#loc-laboratory-protocols

We look forward to receiving your revised manuscript.

Kind regards,

Carlo Vittorio Cannistraci

Academic Editor

PLOS ONE

Additional Editor Comments (if provided):

Dear Authors

We received 3 reviews. All of them suggests major revision.

Please, take care to address carefully all reviewers concerns.

Thanks

Carlo Vittorio Cannistraci

Journal Requirements:

Reviewers' comments:

Reviewer's Responses to Questions

**Comments to the Author**

1. Is the manuscript technically sound, and do the data support the conclusions?

Reviewer #1: Partly

Reviewer #2: Partly

Reviewer #3: Yes

2. Has the statistical analysis been performed appropriately and rigorously? 

Reviewer #1: N/A

Reviewer #2: N/A

Reviewer #3: I Don't Know

3. Have the authors made all data underlying the findings in their manuscript fully available?

Reviewer #1: Yes

Reviewer #2: No

Reviewer #3: Yes

4. Is the manuscript presented in an intelligible fashion and written in standard English?

Reviewer #1: No

Reviewer #2: Yes

Reviewer #3: Yes

5. Review Comments to the Author

Reviewer #1: This is an interesting and novel approach to network alignment. I believe it is a worthwhile contribution to the area, but there are a number of significant aspects and limitations that the authors have completely glossed over or missed.

First and foremost, the authors make the strong claim that they "challenge the notion that topologically similar nodes are functionally related". They present the argument that no network alignment algorithm based solely on network topology has ever shown a significant relationship that nodes aligned using topology alone are functionally similar. While this statement is true, they completely miss the more obvious, and more likely, explanation: existing PPI networks are *grossly* incomplete, and probably quite noisy. While currently available networks show no relationship between network topology and function, it is far too strong a statement to say that there is no relationship. The strongest statement that can be made is that no relationship *has yet been observed*. It is pretty much established canon that we expect topology to be related to function *once we can SEE enough of the topology*. At present, we can't. Just look at BioGRID: human has 330,000 experimentally determined edges among 17,600 nodes; the next most complete mammal is mouse, which has only 20,500 edges between 7200 nodes; followed by rat, at 2700 nodes and 4600 edges. Are you really going to claim that, in three species so *obviously* closely related, that you'd expect to observe similar topology when mouse has at most 30% of its nodes present, and at most 6% of its edges, and rat has only 16% of its nodes and 1.4% of its edges, compared to the closely-related human? Of *course* the GDV vectors are going to be wildly different. When you remove 60% of the nodes and 94% of the edges from mouse, or 85% of nodes and 98.3% edges from rat, of *course* the graphlet counts are going to be wildly different.

Second, it's not at all clear that this method is capable of learning much biology, because if you had, you'd be able to take the topology-function relationships that you learned from one pair of species, and test it on another pair of species. In other words, your argument would be FAR more convincing if you trained your machine on yeast-human, and then tested it on rat-mouse. Instead, you did a 10-fold cross-validation on just one pair of species. In my opinion, all you've done is learned the particular aspects of noise and incompleteness in the current (noisy and incomplete) PPI networks of yeast and human: what happens if you trained your machine on yeast and human BioGRID PPI networks from 2017 (as you appear to have done), and then tested them on the most recent networks from now (November 2019)? I'd be surprised if the results would be as good. In other words, I'm claiming all you've done is learned the noise properties of your particular pair of networks.

If the above was all I had to say, then you might think I'd just say "reject". However, what I find interesting is that your predictions actually do look pretty good. I *still* don't think you can claim to challenge the notion that topological similarity does not imply functional similarity (you MUST prominently mention the noise and incompleteness aspect of current networks, and how such noise is likely MASKING the topological similarity of functionally similar nodes, to make this work publishable), but you *have* found predictive power in your method. I would venture to claim that the reason is as I've stated above: your machine has learned the noise properties of *this particular pair of networks*, and there is predictive information in the properties of these noisy networks that allows you to correlate similar types of noise some functionally related pairs, thereby allowing the transfer of GO terms. This aspect would be MUCH more convincing if you could actually show that you're capable of predicting GO terms.

So, in summary: you have not challenged the notion of topology-function relation at all, you have simply confirmed what is already known: the PPI networks of existing species, even closely related ones like mammals, are all at very disparate levels of completeness. This incompleteness effectively hides the expected topological similarity that should exist between functionally similar nodes (such as homologs), which is why topology-only alignment algorithms have been failing for so long. Your method is successful in learning the *specific* aspects of noise and incompleteness between a *specific* pair of species, allowing you to correlate similar "noisy" topology to similar function. But it's not conniving me that any new biology has been learned, or even *can* be learned, until you can train on one pair of networks, and test on another. Whether the test pair is the name species pair on BioGRID separated by a few years, or a completely different pair of species, is up to you.

A few other important aspects of the paper: this seems like it's been adapted from a thesis. While it would make a good thesis, it is FAR too wordy for a paper submitted to a scholarly journal. By my count, almost half the paper is introduction and previous work. (Pages 1-7, single spaced, before we see Materials and Methods). This is *far* too much introductory material for a paper. It should be about 4-5 TIMES smaller. Everything in the intro could fit into 2-3 well-written pages.

Reviewer #2: In this manuscript Gu and Milenković describe a new method for performing alignment between two graphs. This problem is particularly useful in PPI networks to find homologues between species. The method that they describe uses ML methods to find a node matching predictor using a set of measurable features. The method described seems clever, but it seems there may be some issues with the experimental setup that make it unclear as to the improvement provided by this method. Depending on the resolution to these inquiries this could be a very useful tool to be used in the field.

Major Comments:

* The contact author in the manuscript does not match that in the editorial manager.

* It seems there are some basic assumptions that are not clear. For instance, it seems that there may be many redundant proteins in the network (i.e. performing the exact same function) which would never be able to distinguished. This is not discussed in the experimental measurement methodology in the first section.

* The abstract should contain only a description of the work performed in this manuscript.

* The experimental setup is not well explained and may have flaws. As currently described, it seems that the classifier is trained on part of the input data and tested on another (in the same network). This seems like there would be data leakage from one set to the other. It is not explained why this is not the case.

* Multiple pieces of related work are not cited, in particular the activities of the CAFA group as well as the work related to positive-unlabeled training should be recognized.

* Simulations don't seem to take into account duplicated or deleted genes, this should be justified.

* THe comparisons made on page 9 may not be fair, GO terms are used as the ground truth, but TARA is the only method that utilizes this information in making measurements. At the very least a straw man argument can be made to uses topology and GO similarity to show that the additional overhead of the TARA training mechanism is necessary.

* There are concerns that the data in figure 1(a) does not show any nodes with similarity 1 in the non-matching set. By random chance it would seem that this would still happen unless the dataset is too small.

Minor Comments:

* Terms like "obviously", "clearly", "only" should be avoided. In most cases the places where these clarity assertions are made it is far from clear or are unneeded descriptors.

* "noise" should be replaced with "perturbation"

* The name is quite contrived.

* Only 3 measured are used for comparison while many more methods seem to be described.

* The description of the similarity metrics at line 26 are both too specific (brining in real values) and too general (not adequately descriptive) at the same time.

* On line 9, it is not clear what "complimentary" is supposed to mean, does one confirm the other, are they used on iteration, etc. This may also lead to the discrepancy in the prose in the paragraph starting at line 14, the original NA definition does not include any sequence information while in this section is is discussed differently.

* For the paragraph at line 44, these statements don't necessarily seem untrue for local and multiple. If this is the case, it should be explained.

* The related work section is quite hard to follow and should be made more clear.

* The writing style is very colloquial which somewhat undermines the arguments being made. (see "popular" @ line 4, "here on out" @ line 105)

* The description of the features is not described clearly, it could be improved with a table specifying all of the inputs.

* the disadvantages of the other methods mentioned at line 37 should be explained.

* In general, the figures could be made more clear, instead of simply "similarity measure", the actual metric should be used as the axis label.

Reviewer #3: This paper presents a novel biological network alignment algorithm called TARA. Described as a data driven algorithm, TARA explicitly incorporates (biological) functional node similar metrics by using this data as way to treat NA as a supervised learning problem. More specifically, TARA uses functional similarity to train a binary (logistic regression) classifier to predict which node pairs are functionally similar. From these pairings, TARA is able to produce a network alignment. This approach is contrasted with other as being the first to not assume that topologically similar nodes are functionally related, and provide evidence for why this can be a flawed assumption.

This work takes a step back from treating the problem of biological network alignment as solely one of aligning two graphs, and highlights the fact that the purpose of biological NA is to uncover similarity in biological function. It is an important contribution to this area of research.

While the ideas presented in this paper are novel and insightful, there is much work that can be done in revising its presentation. Generally, the paper requires more structure in its presentation. More specific comments on this:

- The introduction section is very long, and either should use subsections, or some content should be moved out. The experiment in which edges are randomly rewired is out of place and should be moved elsewhere.

- More clearly highlight the difference between “relatedness” and “similarity” as is used in the paper. My understanding is that “similarity” refers specifically to a similarity metric V_1 \\times V_2 \\to R, while “relatedness” refers to the pairing of two nodes. It is important to clearly distinguish these terms, particularly because of sentences such as “…assuming that topological relatedness corresponds to topological similarity”.

- The sentence “so called “percent training” tests” is awkward, as if to imply that this methodology is unsound. Please include some kind of citation or reference to percentage training.

- Provide a more extensive description of the algorithm. For example, psuedocode would be helpful to the reader, even if it is as simple as training a logistic regressor. It is unclear to me how exactly a network alignment is extracted from the results of the regressor. Is the alignment simply all node pairs that are predicted to be positive? Is there a way to extract a one-to-one mapping this way?

6. PLOS authors have the option to publish the peer review history of their article (what does this mean?). If published, this will include your full peer review and any attached files.

Reviewer #1: No

Reviewer #2: No

Reviewer #3: No

---

## [Author Response · Author response to Decision Letter 0]

16 Mar 2020

The authors’ responses to the reviewers’ comments (Please also see the uploaded .docx file designated as "Response to Reviewers")

We would like to thank the Editor and the reviewers for the opportunity to submit the revised version of our manuscript. We believe that the comments we received, along with our changes in response to the comments, have improved the quality of our paper. We believe that we have addressed all of the comments, but we would be happy to make additional modifications if deemed necessary. Below, we list each of the reviewers’ comments, and immediately after we provide our corresponding response, along with describing the corresponding change(s) we implemented in the revised paper. 

Reviewer #1: 

Comment 1: First and foremost, the authors make the strong claim that they "challenge the notion that topologically similar nodes are functionally related". They present the argument that no network alignment algorithm based solely on network topology has ever shown a significant relationship that nodes aligned using topology alone are functionally similar. While this statement is true, they completely miss the more obvious, and more likely, explanation: existing PPI networks are *grossly* incomplete, and probably quite noisy. While currently available networks show no relationship between network topology and function, it is far too strong a statement to say that there is no relationship. The strongest statement that can be made is that no relationship *has yet been observed*. It is pretty much established canon that we expect topology to be related to function *once we can SEE enough of the topology*. At present, we can't. Just look at BioGRID: human has 330,000 experimentally determined edges among 17,600 nodes; the next most complete mammal is mouse, which has only 20,500 edges between 7200 nodes; followed by rat, at 2700 nodes and 4600 edges. Are you really going to claim that, in three species so *obviously* closely related, that you'd expect to observe similar topology when mouse has at most 30% of its nodes present, and at most 6% of its edges, and rat has only 16% of its nodes and 1.4% of its edges, compared to the closely-related human? Of *course* the GDV vectors are going to be wildly different. When you remove 60% of the nodes and 94% of the edges from mouse, or 85% of nodes and 98.3% edges from rat, of *course* the graphlet counts are going to be wildly different.

Response to comment 1: Done. We agree with the reviewer that a likely reason that at present topological similarity does not correspond to functional relatedness is that current PPI network data are noisy. We have now added this discussion in paragraphs 7-9 of Section “Introduction – Background and motivation”. Nonetheless, despite this, NA methods developed to date (including our own!) have still assumed that by optimizing topological similarity they should be able to match functionally related nodes, and our data-driven approach aims to overcome this drawback. So, adding the noise-related discussion actually helps strengthen the need for our approach, and so we thank the reviewer very much for their comment.

We do want to add that we believe that even if/when PPI data become complete, it is still possible if not even likely that topological similarity between functionally related (i.e., evolutionary conserved) PPI network regions will not hold due to biological variation. Namely, molecular evolutionary events like gene or interaction duplication, deletion, or mutation may cause PPI networks to differ across species. Even for protein sequence alignments, pairwise sequence identity as low as 30% is sufficient to indicate evolutionary conservation (i.e., homology) for 90% of all protein pairs. So, one can perhaps expect evolutionary conserved PPI networks of different species to be as topologically dissimilar. We have now also added this discussion to paragraph 10 of Section “Introduction – Background and motivation”.

Comment 2: Second, it's not at all clear that this method is capable of learning much biology, because if you had, you'd be able to take the topology-function relationships that you learned from one pair of species, and test it on another pair of species. In other words, your argument would be FAR more convincing if you trained your machine on yeast-human, and then tested it on rat-mouse. Instead, you did a 10-fold cross-validation on just one pair of species. In my opinion, all you've done is learned the particular aspects of noise and incompleteness in the current (noisy and incomplete) PPI networks of yeast and human: what happens if you trained your machine on yeast and human BioGRID PPI networks from 2017 (as you appear to have done), and then tested them on the most recent networks from now (November 2019)? I'd be surprised if the results would be as good. In other words, I'm claiming all you've done is learned the noise properties of your particular pair of networks.

Response to comment 2: Done. We have now added the particular analysis mentioned by the reviewer – we train TARA on the 2017 yeast and human PPI networks and test the trained TARA on the more recent 2020 versions of the networks. We find that TARA trained on the 2017 networks and tested on the 2020 networks is actually very comparable to TARA trained and tested on the 2017 networks, indicating that TARA is generalizable in this test. For details, see the new section titled “Generalizability of TARA” in Results.

Comment 3: If the above was all I had to say, then you might think I'd just say "reject". However, what I find interesting is that your predictions actually do look pretty good. I *still* don't think you can claim to challenge the notion that topological similarity does not imply functional similarity (you MUST prominently mention the noise and incompleteness aspect of current networks, and how such noise is likely MASKING the topological similarity of functionally similar nodes, to make this work publishable), but you *have* found predictive power in your method. I would venture to claim that the reason is as I've stated above: your machine has learned the noise properties of *this particular pair of networks*, and there is predictive information in the properties of these noisy networks that allows you to correlate similar types of noise some functionally related pairs, thereby allowing the transfer of GO terms. This aspect would be MUCH more convincing if you could actually show that you're capable of predicting GO terms.

Response to comment 3: Respectfully, we believe that we can challenge the notion that topological similarity does not imply functional relatedness – we verify via a set of systematic analyses that this is true. Indeed, as the reviewer notes, this might be true only on the current networks due to data noise, and as requested by the reviewer, we have now added the corresponding discussion (see our response to comment 1 of reviewer 1). However, as we also already responded, biological variation could be another likely cause of topological similarity not implying functional relatedness; if so, the issue will remain even if/when the network data becomes complete. The bottom line is that regardless of the cause, current NA methods fail to capture functionally relatedness as they are all topological similarity-based, and we offer an alternative that can capture functional relatedness better than existing methods. 

Comment 4: So, in summary: you have not challenged the notion of topology-function relation at all, you have simply confirmed what is already known: the PPI networks of existing species, even closely related ones like mammals, are all at very disparate levels of completeness. This incompleteness effectively hides the expected topological similarity that should exist between functionally similar nodes (such as homologs), which is why topology-only alignment algorithms have been failing for so long. Your method is successful in learning the *specific* aspects of noise and incompleteness between a *specific* pair of species, allowing you to correlate similar "noisy" topology to similar function. But it's not conniving me that any new biology has been learned, or even *can* be learned, until you can train on one pair of networks, and test on another. Whether the test pair is the name species pair on BioGRID separated by a few years, or a completely different pair of species, is up to you.

Response to comment 4: Done. See our response to comment 2 of reviewer 1.

Comment 5: A few other important aspects of the paper: this seems like it's been adapted from a thesis. While it would make a good thesis, it is FAR too wordy for a paper submitted to a scholarly journal. By my count, almost half the paper is introduction and previous work. (Pages 1-7, single spaced, before we see Materials and Methods). This isfar* too much introductory material for a paper. It should be about 4-5 TIMES smaller. Everything in the intro could fit into 2-3 well-written pages.

Response to comment 5: Actually, we intentionally had a longer introduction, as we wanted to properly demonstrate drawbacks of the existing methods by showing that (currently) topological similarity does not imply functional relatedness. We do understand that this required several figures to be included in the Introduction, along with description of the underlying analyses. We have now reorganized the paper to address the reviewer’s comment. That is, we have drastically shrunk the Introduction, by moving the results showing that topological similarity does not correlate well with functional relatedness (original Figs. 1 and 2, as well as the corresponding text) to the Section “Results – Topological similarity versus functional relatedness”. We also removed from the Introduction a paragraph that described some methodological details of TARA and was thus redundant to text that already existing in Materials and methods. However, other reviewers requested additional background information that we believe was important to include into the Introduction. In total, the length of the Introduction is somewhat shorter, but not by much. However, the Introduction is now more organized in the light of the reviewer’s comment, and it better highlights the importance and novelties of our method. (Also, please note the very wide right margin of the journal’s paper template.)

Reviewer #2: 

Comment 1: The contact author in the manuscript does not match that in the editorial manager.

Response to comment 1: For us, the contact author in the manuscript is meant to be used regarding correspondence about the manuscript content after publication, while the contact author in the editorial manager is meant to be used regarding correspondence about the submission process. Because different authors handle these tasks, the contacts are different.

Comment 2: It seems there are some basic assumptions that are not clear. For instance, it seems that there may be many redundant proteins in the network (i.e. performing the exact same function) which would never be able to distinguished. This is not discussed in the experimental measurement methodology in the first section.

Response to comment 2: We are not entirely sure we understand this comment. If the reviewer means that different proteins can be annotated by the same GO term, indeed this is the case, and this is a common knowledge. We do not understand why this is perceived a problem, however. Our goal (and goal of any other protein function prediction study) is to use some proteins for which we know their function(s), i.e., their shared GO term(s) to identify which other proteins have the same function(s). No protein functional prediction can be done (by us on anyone else) unless significantly sufficiently many proteins perform the same function(s). In our eyes, proteins that share GO term(s) should not be considered as redundant proteins; they should simply be considered as a part of the same functional module that carries out the particular shared function(s). If the reviewer is not satisfied with our explanation, has a follow-up question, or has a specific suggestion on what change(s) to make in our paper, we would be happy to address that. 

Comment 3: The abstract should contain only a description of the work performed in this manuscript.

Response to comment 3: Done. We have now rewritten relevant parts of the abstract to focus only on the work done in our study.

Comment 4: The experimental setup is not well explained and may have flaws. As currently described, it seems that the classifier is trained on part of the input data and tested on another (in the same network). This seems like there would be data leakage from one set to the other. It is not explained why this is not the case.

Response to comment 4: Fixed. We have added three pieces of pseudocode, Algorithms 1, 2 and 3, to better explain the experimental setup. Just like with any supervised classification study, it is indeed the case that the TARA classifier is trained on one part of the input data (i.e., on functional labels of a subset of node pairs across the compared network) and tested on a different, independent, yet-unseen part of the input data (i.e., on functional labels of a different subset of node pairs). So, we are not sure why the reviewer suspects this to be a problem, as this is a standard machine learning evaluation. Also, we are not sure why the reviewer believes that there could be data leakage from the training set to the testing set, as the two sets of node pairs do not overlap. 

Comment 5: Multiple pieces of related work are not cited, in particular the activities of the CAFA group as well as the work related to positive-unlabeled training should be recognized.

Response to comment 5: Done. We have now included discussion on additional work, including CAFA, in the last paragraph of Section “Introduction -- Related work”.

Comment 6: Simulations don't seem to take into account duplicated or deleted genes, this should be justified.

Response to comment 6: We are not entirely sure we understand this comment. If the reviewer means that the network data may contain duplicated or deleted entries, we use publicly available data from BioGRID where each gene has a unique ID, and thus no node is duplicated in or deleted from our network. If the reviewer might be referring to evolutionary events of gene duplication or deletion, such events are a key motivation behind proposing our data-driven, i.e., topological relatedness-based, NA framework as an alternative to the existing topological similarity-based NA methods. For details, see paragraph 10 of Section “Introduction – Background and motivation”, paragraph 2 of Section “Introduction – Our contributions”, and newly added Fig. 1. 

Comment 7: The comparisons made on page 9 may not be fair, GO terms are used as the ground truth, but TARA is the only method that utilizes this information in making measurements. At the very least a straw man argument can be made to uses topology and GO similarity to show that the additional overhead of the TARA training mechanism is necessary.

Response to comment 7: TARA is indeed the only considered NA method using GO data to guide the alignment process in a supervised manner. This is because TARA is the only existing (considered or non-considered) supervised NA method that uses GO data. In other words, none of the methods we compare against can use topology and GO similarity; the incorporation of GO data into the alignment process is exactly one of the novel contributions of our method. If we were to modify an existing method to also use GO data, the resulting method would be a novelty of our study. That is actually what we have done in the original paper – we have learned from the best existing NA methods (WAVE, SANA, and PrimAlign) to develop a supervised method that can use GO term data. Then, one of our goals has been to show that using both topological and functional information (via our TARA approach) outperforms using only topological information (via existing WAVE, SANA, and PrimAlign approaches). This is similar to previous NA methods showing that using both topological and protein sequence information outperforms using only topological information. Note that using all three (topology, sequence, and function), which we expect to outperform using any combination of two, is the subject of our on-going and future current work that is out of the scope of the current papers.

Comment 8: There are concerns that the data in figure 1(a) does not show any nodes with similarity 1 in the non-matching set. By random chance it would seem that this would still happen unless the dataset is too small.

Response to comment 8: In 1(a) (now 2(a) in the revised paper), less than 1% of node pairs are in the final similarity bin (0.95 to 1) in the non-matching set. The scale only makes it appear that there are none, but there are some, it is just that they are not visible.

Comment 9: Terms like "obviously", "clearly", "only" should be avoided. In most cases the places where these clarity assertions are made it is far from clear or are unneeded descriptors.

Response to comment 9: Done. We have removed any use of “obviously” or “clearly”, and we now use “only” when absolutely necessary.

Comment 10: "noise" should be replaced with "perturbation"

Response to comment 10: Done throughout the paper. 

Comment 11: The name is quite contrived.

Response to comment 11: True. However, while the name of our method is not formed from the initials of "data-driven network alignment", the purpose of our naming is to have an easy-to-say method, which TARA accomplishes. Choosing a method name is a highly subjective decision, and we as the authors strongly prefer to continue using TARA.

Comment 12: Only 3 measured are used for comparison while many more methods seem to be described.

Response to comment 12: We believe that by “measured”, the reviewer meant “existing NA methods”. If so, in Section "Introduction -- Related work", we discuss why we only compare against WAVE, SANA, and PrimAlign. Namely, the other listed NA methods were all outperformed by WAVE, SANA, and PrimAlign on the exact same data. Therefore, we can transitively compare our method with any mentioned, which is why we believe the three we consider are sufficient.

Comment 13: The description of the similarity metrics at line 26 are both too specific (brining in real values) and too general (not adequately descriptive) at the same time.

Response to comment 13: Done. We have clarified the description of topological similarity, including bringing all of the discussion to the same (consistent) level.

Comment 14: On line 9, it is not clear what "complimentary" is supposed to mean, does one confirm the other, are they used on iteration, etc. This may also lead to the discrepancy in the prose in the paragraph starting at line 14, the original NA definition does not include any sequence information while in this section is is discussed differently.

Response to comment 14: Done. We have clarified what we mean by complementary. Specifically, NA can be used to predict what sequence alone cannot.

Comment 15: For the paragraph at line 44, these statements don't necessarily seem untrue for local and multiple. If this is the case, it should be explained.

Response to comment 15: Done. We have rewritten the paragraph such that the statements apply to global and pairwise as well as local and multiple NA.

Comment 16: The related work section is quite hard to follow and should be made more clear.

Response to comment 16: Done. We have made minor changes in the writing and major changes in organization of that section to make it easier to follow. Prominently, we broke text into smaller paragraphs so that the logical train of thoughts is now easier to follow. 

Comment 17: The writing style is very colloquial which somewhat undermines the arguments being made. (see "popular" @ line 4, "here on out" @ line 105)

Response to comment 17: Done. We have rewritten relevant parts of the paper to be less colloquial.

Comment 18: The description of the features is not described clearly, it could be improved with a table specifying all of the inputs.

Response to comment 18: Done. We have added three pieces of pseudocode, Algorithms 1, 2 and 3, clarifying how the features are calculated and how they are inputted into our framework.

Comment 19: the disadvantages of the other methods mentioned at line 37 should be explained.

Response to comment 19: Done. We have elaborated on the advantages and disadvantages of local and global NA.

Comment 20: In general, the figures could be made more clear, instead of simply "similarity measure", the actual metric should be used as the axis label.

Response to comment 20: Done. We have clarified the figures.

Reviewer #3: 

Comment 1: The introduction section is very long, and either should use subsections, or some content should be moved out. The experiment in which edges are randomly rewired is out of place and should be moved elsewhere.

Response to comment 1: Done. We have added subsections to the Introduction (Background and motivation, Our contributions) and moved the original Figs. 1 and 2 and the corresponding text to the Results section. We have also removed from Introduction a paragraph that described some methodological details of TARA, as it was redundant to text that already existed in Materials and methods.

Comment 2: More clearly highlight the difference between “relatedness” and “similarity” as is used in the paper. My understanding is that “similarity” refers specifically to a similarity metric V_1 \\times V_2 \\to R, while “relatedness” refers to the pairing of two nodes. It is important to clearly distinguish these terms, particularly because of sentences such as “…assuming that topological relatedness corresponds to topological similarity”.

Response to comment 2: Done. We have added a new figure (Fig. 1 in the revised paper) illustrating the difference between similarity and relatedness, along with the corresponding explanation of the figure in paragraph 2 of Section “Introduction – Our contributions”. 

Comment 3: The sentence “so called “percent training” tests” is awkward, as if to imply that this methodology is unsound. Please include some kind of citation or reference to percentage training.

Response to comment 3: Done. We have reworded this to be more clear. We meant the phrase to indicate that we call the tests "percent training" in this paper for simplicity when referring to them, but the tests are a commonly adopted and sound practice when evaluating a supervised classifier.

Comment 4: Provide a more extensive description of the algorithm. For example, psuedocode would be helpful to the reader, even if it is as simple as training a logistic regressor. It is unclear to me how exactly a network alignment is extracted from the results of the regressor. Is the alignment simply all node pairs that are predicted to be positive? Is there a way to extract a one-to-one mapping this way?

Response to comment 4: Done. We have added three pieces of pseudocode, Algorithms 1, 2, and 3, outlining how we obtain the feature vector of a node pair, how we create a balanced dataset, and how we generate an alignment. We hope that the new details now allow for understanding our methodology and analysis at the level of reproducibility. 

In our current framework, TARA outputs a many-to-many alignment, and there is no way to guarantee that a one-to-one alignment can be found; this is subject of our future work.

---

## [Decision Letter · Decision Letter 1]

8 Jun 2020

Data-driven network alignment

PONE-D-19-28710R1

Dear Dr. Gu,

We’re pleased to inform you that your manuscript has been judged scientifically suitable for publication and will be formally accepted for publication once it meets all outstanding technical requirements.

Kind regards,

Carlo Vittorio Cannistraci

Academic Editor

PLOS ONE

Additional Editor Comments (optional):

Congratulations the article is accapted!

Reviewers' comments:

Reviewer's Responses to Questions

**Comments to the Author**

1. If the authors have adequately addressed your comments raised in a previous round of review and you feel that this manuscript is now acceptable for publication, you may indicate that here to bypass the “Comments to the Author” section, enter your conflict of interest statement in the “Confidential to Editor” section, and submit your "Accept" recommendation.

Reviewer #1: All comments have been addressed

Reviewer #2: (No Response)

2. Is the manuscript technically sound, and do the data support the conclusions?

Reviewer #1: Yes

Reviewer #2: Partly

3. Has the statistical analysis been performed appropriately and rigorously? 

Reviewer #1: Yes

Reviewer #2: Yes

4. Have the authors made all data underlying the findings in their manuscript fully available?

Reviewer #1: Yes

Reviewer #2: Yes

5. Is the manuscript presented in an intelligible fashion and written in standard English?

Reviewer #1: Yes

Reviewer #2: Yes

6. Review Comments to the Author

Reviewer #1: All of my previous comments have been adequately addressed. While I still do not believe the truth of the claim that topology and function are unrelated, the authors have done an adequate job of answering all my previous criticisms, and an admirable job of presenting their evidence more clearly. My belief in the truth or falsity of the claim is irrelevant; the authors have done a decent job of carefully presenting their case and the evidence supporting their claim.

Reviewer #2: The authors have done work to improve this paper and have addressed most of the reivewers comments.

I agree with the other reviewers that the claims of the papers findings are overstated, I think they could be toned down even beyond the level in the edited manuscript.

7. PLOS authors have the option to publish the peer review history of their article (what does this mean?). If published, this will include your full peer review and any attached files.

Reviewer #1: No

Reviewer #2: No

---

## [Editor Report · Acceptance letter]

17 Jun 2020

PONE-D-19-28710R1 

Data-driven network alignment 

Dear Dr. Gu:

I'm pleased to inform you that your manuscript has been deemed suitable for publication in PLOS ONE. Congratulations! Your manuscript is now with our production department. 

Kind regards, 

on behalf of

Dr. Carlo Vittorio Cannistraci 

Academic Editor

PLOS ONE